# EXPRESSIVE GRAPH NEURAL NETWORKS VIA EQUIVARIANT USE OF NOISE

## ABSTRACT

Expressivity has been a major focus in the design of Graph Neural Networks (GNNs), yet a significant gap persists between theoretical universal expressivity and practical performance. While many expressive GNNs are efficient and achieve strong results, they often focus on specific graph properties and lack theoretical expressivity for general graph tasks. Conversely, theoretically universal-expressive models often suffer from high computational costs or poor generalization, limiting their real-world applicability. To bridge this gap, we introduce Equivariant Noise GNNs (ENGNNs), a framework that utilizes random noise features to enhance the expressivity of GNNs. Crucially, unlike prior methods that naively use noise, we enforce equivariance to nodewise noise transformations, such as orthogonal transformations. We prove that this property reduces the model's theoretical sample complexity, thereby improving generalization. Our framework simultaneously reaches theoretical universal expressivity, maintains the linear scalability of standard Message-Passing Neural Networks in practice, and achieves performance comparable to computationally expensive, high-expressivity models. Extensive experiments confirm strong performance across node, link, subgraph, and graph-level prediction tasks, demonstrating that the equivariant use of noise provides a powerful and practical pathway for building expressive GNNs. Our code is available at https://anonymous.4open.science/r/EquivNoiseGNN/.

## 1 INTRODUCTION

Graph Neural Networks (GNNs) have emerged as powerful tools for graph representation learning, with applications in areas such as natural language processing (Yao et al., 2019), bioinformatics (Fout et al., 2017), and social network analysis (Chen et al., 2018). However, popular architectures like Message Passing Neural Networks (MPNNs) (Gilmer et al., 2017) face fundamental expressivity limitations, hindering their performance on complex node-, link-, and graph-level tasks (Dwivedi et al., 2022b; Li et al., 2018; Zhang & Chen, 2018; Zhang et al., 2021). Consequently, enhancing the expressivity of GNNs has become a central focus.

Research on expressivity generally follows two paths: (1) improving general capacity for graph isomorphism testing and function approximation, as seen in high-order GNNs (Morris et al., 2019; Maron et al., 2019a;b), and (2) designing models to express specific graph properties relevant to a particular task, such as methods for path/neighborhood overlap between nodes (Zhu et al., 2021b; Chamberlain et al., 2023a). Path (2) is often limited to specific tasks. Path (1) is task-agnostic but has seen limited practical adoption. Among approaches for general expressivity, augmenting nodes with random noise features is theoretically appealing. It provides a task-agnostic mechanism to make nodes distinguishable, provably reaching universal expressivity (Abboud et al., 2021). However, this method has seen limited practical usage. The core issue is that naively using noise dramatically increases the model's input space, leading to poor generalization. Early works (Sato et al., 2021; Abboud et al., 2021) verify noise's effectiveness on synthetic datasets where expressivity is paramount, but on real-world tasks, the generalization error caused by the noise often outweighs its benefits.

To overcome this generalization challenge, we propose to enforce symmetry in the noise space. We argue that while the noise itself should be random to break graph symmetries, the function processing the noise should respect graph symmetries in general. Our key insight is that by making a GNN **invariant to a group of transformations applied to each node's noise** (e.g., orthogonal

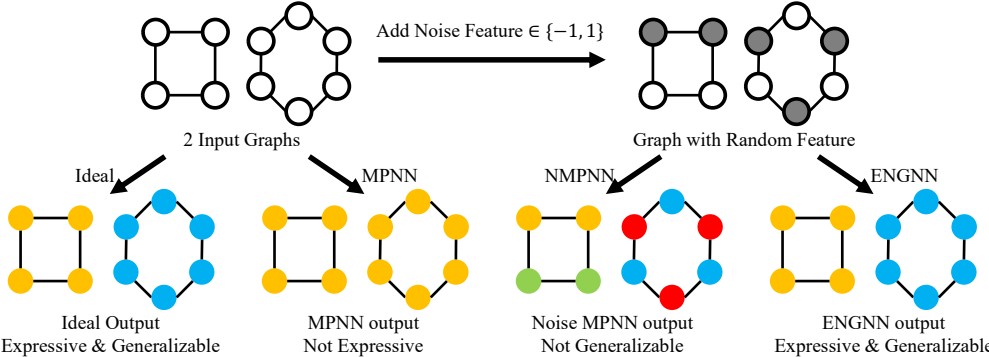

Figure 1: We compare vanilla MPNNs, Noise MPNNs (NMPNN), and our Equivariant Noise GNNs (ENGNN) using a 4-cycle and a 6-cycle as input, with node representations indicated by color. Ideal output should distinguish two cycles while assigning identical representations to symmetric nodes within the same cycle. MPNNs will produce the same representation to all nodes, failing to differentiate two cycles. NMPNN improves expressivity by introducing noise features (e.g., a 1D random variable in -1, 1) but compromises generalization by differentiating symmetric nodes in the same cycle. In contrast, ENGNN processes noise features equivariantly to predefined transformations (e.g. 1D orthogonal transformations), enabling it to differentiate two cycles and produce same output for symmetric nodes, thus achieving both high expressivity and better generalization than NMPNN.

transformations or channel permutations), we can drastically reduce the sample complexity bound. As illustrated in Figure 1, this allows the model to use noise to distinguish non-isomorphic graphs (like a 4-cycle and a 6-cycle) while correctly assigning identical representations to symmetric nodes within a single graph—a property that naive noise models struggle to maintain.

Our solution, the Equivariant Noise GNN (ENGNN), achieves invariance through a two-stream architecture. An invariant stream, initialized with standard node features, is processed alongside an equivariant stream, initialized with random noise. A specially designed aggregator layer, which is equivariant to a chosen set of noise transformations, then mixes information from these two streams while preserving their symmetry properties throughout the network. The final prediction relies on invariant pooling results, maintaining the output's invariance to the chosen noise transformations.

Theoretically, we establish two results: 1) We prove that ENGNNs are provably invariant to the chosen noise transformations, leading to a tighter generalization bound compared to naive noise methods. 2) We prove that, with a universally expressive aggregator, ENGNNs achieve universal expressivity not just for graph-level tasks, but also for node-, link-, and subgraph-level predictions, significantly broadening their applicability beyond the theory of prior noise-based GNNs.

We implement two variants, ENGNN-O (equivariant to orthogonal transformations) and ENGNN-P (equivariant to permutation), to validate our method. Experiments demonstrate that ENGNNs consistently outperform both naive MPNNs and Noise MPNNs across all tasks. Furthermore, ENGNNs achieve performance comparable to highly expressive but computationally expensive models (e.g., subgraph GNNs, high-order GNNs) while maintaining the linear time and space complexity of a standard MPNN in practice. This combination of scalability, expressivity, and versatility positions ENGNN as a powerful and practical successor to MPNNs for real-world graph learning.

## 2 RELATED WORK

**Expressive GNNs.** Building expressive GNNs is challenging due to the intricate graph topology and the permutation invariance requirement. Research has largely split into two directions.

The first path aims for theoretical universal expressivity: creating models that can, in principle, distinguish any two non-isomorphic graphs. These methods often draw inspiration from the Weisfeiler-Lehman (WL) test (Xu et al., 2019; Maron et al., 2019a; Morris et al., 2019; Zhang et al., 2023; Bevilacqua et al., 2022; Qian et al., 2022; Zhou et al., 2023a). Others focus on constructing function approximators for graphs, like layer layers and polynomials, using high-order tensors (Maron et al., 2019b; Puny et al., 2023; Frasca et al., 2022). While theoretically powerful, these approaches often come with a high computation cost, as they rely on processing high-order tensors or sampling a large number of subgraphs (Zhang & Li, 2021; Huang et al., 2023b; Zhao et al., 2022), making them

impractical for node, link, or even moderately-sized graph-level tasks. Besides high-order GNNs, another direction relaxes the strict requirement for permutation invariance by assigning nodes additional features. The relational pooling method (Murphy et al., 2019) assigns unique node IDs and pools results over different permutations, but enumerating all permutations is computationally expensive, so only a random subset is used in practice. Other methods assign nodes random noise vectors (Abboud et al., 2021; Sato et al., 2021) or integer features from heuristics (Dasoulas et al., 2020; Franks et al., 2023; Pellizzoni et al., 2024; Garg et al., 2020). While these approaches achieve universal expressivity, they often suffer from poor generalization and are not widely applied.

The second path prioritizes specific tasks over universal approximation. These models are designed to capture specific graph properties relevant to a domain. Examples include GNNs for subgraph counting in molecules (Chen et al., 2020; Huang et al., 2023b), spectral GNNs that mimic graph filters (Wang & Zhang, 2022b; Defferrard et al., 2016; He et al., 2021; Chien et al., 2021; Klicpera et al., 2019), Graph Transformers for capturing long-range dependencies (Mialon et al., 2021; Kreuzer et al., 2021; Ying et al., 2021; Rampásek et al., 2022), positional encoding methods for improving whole graph task (Dwivedi et al., 2022a; Beaini et al., 2021; Huang et al., 2024; Wang et al., 2022; Lim et al., 2023; Huang et al., 2023a; Ma et al., 2023; Li et al., 2020), and link prediction models that leverage path information (Wang et al., 2024; Chamberlain et al., 2023a; Zhang & Chen, 2018; Yun et al., 2021; Zhu et al., 2021b). While highly effective in their target domains, these methods lack the theoretical expressivity for general graph problems.

Our work, ENGNN, bridges the gap between theoretical expressivity and real-world applicability. It propose equivariant use of noise to achieve universal expressivity without resorting to computationally expensive high-order operations, making it both powerful and broadly applicable.

**GNNs with Noise.**  A promising strategy for increasing expressivity at a low cost is to augment nodes with random features, effectively breaking symmetries that MPNNs cannot. Initial works simply concatenate random vectors (e.g., from Gaussian or uniform distributions) to node features (Abboud et al., 2021; Sato et al., 2021). This is shown to achieve universal expressivity in theory but suffers from poor generalization in practice. Resampling noise during training was proposed as a mitigation strategy (Abboud et al., 2021), but it failed to fully resolve the underlying generalization issue. Subsequent works used noise more cautiously, typically as a tool to approximate a specific, permutation-invariant heuristic. For example, MPLP (Dong et al., 2024) uses random projections to approximate common neighbor statistics, while others use noise GNNs to fit Laplacian eigenvectors or random walk encodings (Cantürk et al., 2024; Franks et al., 2025; Eliasof et al., 2023). These methods improve generalization by collapsing the noise into fixed heuristics, but they sacrifice the universal expressivity that made noise theoretical appealing. In contrast, ENGNN enforces equivariance on noise, controlling the model's sample complexity without losing the universal expressivity required for general graph tasks. Besides these works on continuous random vector features, some works (Murphy et al., 2019; Dasoulas et al., 2020) that assign random integers to nodes can also be considered as noise GNNs. Subsequent works (Franks et al., 2023; Pellizzoni et al., 2024; Garg et al., 2020) proposed assigning the same integer to some nodes to reduce sample complexity, but these works still focus on theory rather than application to broad graph tasks.

**Equivariant Graph Neural Networks.**  Equivariance is ensuring that a model's output transforms predictably with transformations of its input. In GNN domain, this principle has been widely applied to physical data like 3D molecules or point clouds. Models like TFN, EGNN, and PaiNN (Thomas et al., 2018; Satorras et al., 2021; Schütt et al., 2021; Dym & Maron, 2021; Batzner et al., 2022; Shi & Rajkumar, 2020) are designed to be equivariant to rotations and translations of the input coordinates. Here, the goal is to respect the intrinsic physical symmetries of the input data. While Satorras et al. (2021) also explore invariance in a Graph Autoencoder's latent space, the primary focus of previous equivariant GNNs remains on physical symmetries. In contrast, our ENGNN's goal is not to preserve a physical property of the input, but rather to improve the model itself. Our goal is not to develop new equivariant network, but to apply equivariant method to noise.

## 3 PRELIMINARIES

For a matrix $Z \in \mathbb{R}^{a \times b}$, let $Z_i \in \mathbb{R}^b$ denote the $i$-th row (as a column vector), $Z_{:,j} \in \mathbb{R}^a$ denote the $j$-th column, and $Z_{ij} \in \mathbb{R}$ denote the element at the $(i,j)$-th position. A *graph* is represented as

$G = (V, E, X)$, where $V = 1, 2, 3, \ldots, n$ is the set of $n$ nodes, $E \subseteq V \times V$ is the set of edges, and $X \in \mathbb{R}^{n \times d}$ is the node feature matrix, with the $v$-th row $X_v$ representing the features of node $v$. $E$ can be expressed with the adjacency matrix $A \in \mathbb{R}^{n \times n}$, where $A_{uv} = 1$ if the edge $(u, v) \in E$, and 0 otherwise. A graph $G$ can be simply denoted by the tuple $(A, X)$. Let $\mathcal{G}$ denote the graph space.

**Graph Isomorphism.** A graph's structure is independent of the ordering of its nodes. This concept is formalized by *graph isomorphism*. Two graphs, $G_1 = (A_1, X_1)$ and $G_2 = (A_2, X_2)$, are *isomorphic* if there exists a *permutation* matrix $P$ such that $PA_1P^T = A_2$ and $PX_1 = X_2$.

**Message Passing Neural Network (MPNN) (Gilmer et al., 2017).** MPNN is a popular GNN framework. It consists of multiple message-passing layers, where the $k$-th layer is:

$$\boldsymbol{h}_v^{(k)} = U^{(k)}(\boldsymbol{h}_v^{(k-1)}, \mathrm{AGG}(\{M^{(k)}(\boldsymbol{h}_u^{(k-1)}) \mid u \in V, (u, v) \in E\})), \tag{1}$$

where $\boldsymbol{h}_v^{(k)}$ is the representation of node $v$ at the $k$-th layer, $U^{(k)}$ and $M^{(k)}$ are functions such as Multi-Layer Perceptrons (MLPs), and AGG is an aggregation function like sum or max. The initial node representation $\boldsymbol{h}_v^{(0)}$ is the node feature $X_v$. Each layer aggregates information from neighbors to update the center node's representation.

**Equivariance and Invariance.** Given a function $h : \mathcal{X} \to \mathcal{Y}$ and a group of operators $T$ acting on $\mathcal{X}$ and $\mathcal{Y}$ through operation $\star$, $h$ is $T$-*invariant* if $h(t \star x) = h(x), \quad \forall x \in \mathcal{X}, t \in T$, and $T$-*equivariant* if $h(t \star x) = t \star h(x), \quad \forall x \in \mathcal{X}, t \in T$. For example, graph-level tasks require models invariant to node order permuation as most graph properties are invariant to the ordering of nodes, and equivariant use of noise keep equivariance to nodewise transformation on noise of each node.

**Universal Expressivity.** Following Chen et al. (2019), a GNN framework $f$ is considered *universally expressive* if it is *Graph-Isomorphism-discriminating*, meaning for any two non-isomorphic graphs $G_1, G_2$, there exists a parameterization $\theta$ such that the GNN outputs are different: $f_\theta(G_1) \neq f_\theta(G_2)$. This is equivalent to being able to approximate any continuous, permutation-invariant function on graphs. We adopt this definition to prove the expressivity of our model.

## 4 EQUIVARIANT NOISE GRAPH NEURAL NETWORK (ENGNN)

This section introduces our Equivariant Noise Graph Neural Network (ENGNN). The core idea is to use random noise to achieve universal expressivity while leveraging the principle of equivariance to mitigate the poor generalization typically caused by naive noise injection. The framework can be adapted to be invariant to different noise transformations via specialized aggregators (e.g., orthogonal or permutation-equivariant aggregators in Appendix G).

**Key Notations.** The input graph is $G$ with $n$ nodes and $m$ edges, with auxiliary noise $Z \in \mathcal{Z}$, where $\mathcal{Z} = \mathbb{R}^{n \times C}$ and each node $i$ has a noise vector in $\mathbb{R}^C$. We consider a nodewise transformation group $T$ acting on the noise space, where each element $t \in T$ is a function $t : \mathbb{R}^C \to \mathbb{R}^C$. The transformation $t$ is applied to $Z$ row-wise, so that $t(Z)_i = t(Z_i)$. We consider functions that are equivariant or invariant to this group of noise transformations $T$.

### 4.1 EQUIVARIANCE HELPS GENERALIZATION

As shown in Figure 1, noise can break the inductive bias that GNNs should produce the same representations for symmetric nodes, leading to poor generalization. In contrast, equivariant noise GNNs reduce such cases, leading to better generalization. This section provides a theoretical explanation based on sample complexity from PAC (Probably Approximately Correct) learning theory. An introduction to PAC learning theory, related work on generalization, and proofs are in Appendix B. Note that previous works have analyzed GNN generalization bounds, but our results explain the benefit of noise equivariance in our ENGNN, which previous results cannot apply directly.

We first define common settings in PAC learning theory: The prediction target lies in a compact set $\mathcal{Y} \subseteq \mathbb{R}^d$. The hypothesis class $H$ consists of functions $h : \mathcal{G} \times \mathcal{Z} \to \mathcal{Y}$ mapping graph-noise pairs to predictions. The loss function $l(y, y')$ is bounded and Lipschitz continuous with constant $C_l$.

We introduce the covering number concept and show its connection to symmetry. Intuitively, if the model is invariant to $T$, different noise points that can be transformed into each other are mapped to the same output. Therefore, the transformations can reduce the effective distance between noise

data points without changing their outputs. Let $\rho_Z$ denote a metric (e.g., Euclidean distance) on the noise space $\mathcal{Z}$. The *semi-metric on the noise space induced by $T$* is:

$$\rho_{Z,T}(Z_1, Z_2) = \inf_{t_1, t_2 \in T} \rho_Z(t_1(Z_1), t_2(Z_2)), \tag{2}$$

which reduces noise distances by finding and applying transformations. The covering number for the noise space $N(\mathcal{Z}, \rho_{Z,T}, r)$ is the minimum number of points needed so that every point in $\mathcal{Z}$ is within a distance $r$ (under the semi-metric) of one of these points.

The following theorem provides a sample complexity bound for $T$-invariant models:

**Theorem 4.1.** *Assume all $h \in H$ are $C_G$-Lipschitz in $\mathcal{G}$, $C_Z$-Lipschitz in $\mathcal{Z}$, and $T$-invariant, and the loss function is $C_l$-Lipschitz. The sample complexity for empirical risk minimization is:*

$$O\left(\frac{1}{\epsilon^2} N_{Z,T} N_G \ln N_Y + \frac{1}{\epsilon^2} \ln \frac{1}{\delta}\right), \tag{3}$$

*where 1) $\epsilon, \delta$ are the error bound and failure probability, 2) $N_Z = N(\mathcal{Z}, \rho_{Z,T}, \frac{\delta}{12 C_l C_Z})$ is the covering number for noise space $\mathcal{Z}$ with semi-metric $\rho_{Z,T}$ induced by $T$ and radius $\frac{\delta}{12 C_l C_Z}$, 3) $N_G$ and $N_Y$ are covering numbers for graph space $G$ and output space $\mathcal{Y}$. They are irrelevant to $T$.*

The only term related to $T$ is $N_{Z,T}$, the covering number of the noise space. For a naive Noise MPNN (which is only invariant to the identity transformation), the covering number can be enormous. For an $n \times C$-dimensional boolean noise space, $N_Z$ can be as large as $2^{nC}$, growing exponentially with the number of nodes and noise channels. This explains why noise leads to poor generalization. Incorporating symmetries in the noise space can reduce sample complexity. As more symmetries involves (corresponding to a larger $T$), points are closer to one another under the semi-metric $\rho_{Z,T}$. This leads to a smaller covering number:

**Proposition 4.2.** *If $T_1 \subseteq T_2$, then for all $r > 0$, $N(\mathcal{Z}, \rho_{Z,T_1}, r) \geq N(\mathcal{Z}, \rho_{Z,T_2}, r)$.*

Moreover, this reduction can be dramatic. For example, by enforcing invariance to the permutation of noise channels, we can reduce the covering number by a factorial factor:

**Proposition 4.3.** *If $\mathcal{Z} = [0,1]^{n \times C}$, and $T$ includes all permutations of the $C$ noise channels, then for a small enough radius $r$: $N(\mathcal{Z}, \rho_{Z,T}, r)/N(\mathcal{Z}, \rho_Z, r) \leq 2/C!$.*

This theoretical framework shows that a principled application of invariance is key to harnessing the expressive power of noise without suffering from poor generalization. Other methods, like simply reducing the noise dimension, can hurt expressivity by causing collisions (distinct nodes receiving similar noise vectors, as shown in Appendix H and C), making invariance the superior approach.

## 4.2 ARCHITECTURE

The ENGNN architecture is designed to process information while respecting these noise symmetries. Let $d$ and $L$ denote hidden dimensions, and $C$ denote noise channels. Each node $i$ maintains two representations that are updated across $K$ message-passing layers. At $k$-th layer:

- Invariant representation $X_i^{(k)} \in \mathbb{R}^d$, which remains unchanged under noise transformations.
- Equivariant representation $Z_i^{(k)} \in \mathbb{R}^{L \times C}$, which transforms as the input noise $Z_i^{(0)} \in \mathbb{R}^C$.

Initialized with noise and node features, ENGNN updates both via equivariant MPNN layers.

**Equivariant Aggregator.** Equivariant Aggregator takes a multi-set of invariant-equivariant feature pairs $\{(X_i^{(k)}, Z_i^{(k)}) | i = 1, 2, ..., B\}$ to produce a feature pair $(X', Z')$. We use AGGR to represent an aggregator. Design of aggregators equivariant to different transformation sets is in Appendix G. They achieve equivariance to input, theoretical universal expressivity under mild condition, and linear time and space complexity to input set size in practice.

**Message-Passing Layer.** Each node $i$ updates its representations as follows:

$$X_i^{(k)}, Z_i^{(k)} = \text{AGGR}_1^{(k)}\left(\left\{\left(\text{AGGR}_2^{(k)}(\{(X_j^{(k-1)}, Z_j^{(k-1)}) \mid j \in N(i)\}, \text{MLP}^{(k)}(X_i^{(k-1)}), Z_i^{(k-1)}))\right)\right\}\right), \tag{4}$$

where $\text{AGGR}_2^{(k)}$ aggregates neighbors' feature, $\text{AGGR}_1^{(k)}$ combines aggregated features with the center node's feature, and $\text{MLP}^{(k)}$ transforms center node's feature to distinguish it from neighbors'.

**Pooling Layer.** To generate graph-level representations, we aggregate all nodes:

$$h_G, Z_G = \text{AGGR}(\{(X_i^{(K)}, Z_i^{(K)}) \mid i \in V\}), \tag{5}$$

where the invariant output $h_G$ is used as the graph representation for downstream tasks.

For tasks involving nodes, links, or subgraphs, representations for a node subset $U \subseteq V$ is:

$$h'_U, Z'_U = \text{AGGR}_1(\{(X_i^{(K)}, Z_i^{(K)}) \mid i \in U\}), \tag{6}$$

$$h_U, Z_U = \text{AGGR}_2(\{(\text{MLP}(h_G), Z_G), (h'_U, Z'_U)\}), \tag{7}$$

where $h'_U, Z'_U$ aggregates subset node features, and $\text{AGGR}_2$ combines subset feature and global feature $h_G, Z_G$ leads to the final subgraph representations $h_U$.

**Complexity.** With efficient aggregators (Appendix G) that scale linearly with input size, ENGNN achieves $O(n+m)$ time and space complexity per message-passing layer, where $n$ is the number of nodes and $m$ the number of edges. The pooling step costs $O(n)$ time and space for full graphs and $O(n + \sum_i^B |U_i|)$ time and space for $B$ node subsets $U_1, U_2, \ldots, U_B$. Therefore, ENGNN maintains the same scalability as vanilla MPNNs and scales much better that high-order GNNs. **Note that to retain theoretical universal expressivity, large depth and width depending on the task may be needed, which can make the theoretical time and space complexity non-linear with respect to graph size. However, in experiments, only modest depth and width are sufficient for strong empirical performance, leading to linear time and space complexity in practice.**

### 4.3 THEORETICAL EXPRESSIVITY

All proofs in this section are in Appendix D. First, ENGNN inherits the equivariance of its aggregator and ensures output's invariance to noise transformations:

**Theorem 4.4.** *(Invariance) If the aggregator is equivariant to a group of nodewise noise transformations $T$, then ENGNN's outputs for graphs and subsets are invariant to $T$.*

Second, ENGNN can achieve universal expressivity in both graph and subgraph tasks:

**Theorem 4.5.** *Assume each node has distinct random features, and the aggregator in ENGNN achieves universal expressivity with these noise features (e.g. Aggregators in Appendix G).*

*(Graph-Level Expressivity) Let ENGNN$(G, Z)$ denote the model's output for graph $G$ and noise $Z$. For any non-isomorphic graphs $G$ and $H$, there exists a parameterization such that:*

$$ENGNN(G, Z_1) \neq ENGNN(H, Z_2), \quad \forall Z_1, Z_2 \in \mathcal{Z}. \tag{8}$$

*(Subgraph-Level Expressivity) Let ENGNN$(U, G, Z)$ denote the output for subset $U$ in graph $G$. For any non-isomorphic pairs $(G, U_G)$ and $(H, U_H)$, there exists a parameterization such that:*

$$ENGNN(U_G, G, Z_1) \neq ENGNN(U_H, H, Z_2), \quad \forall Z_1, Z_2 \in \mathcal{Z}. \tag{9}$$

These results demonstrate that ENGNN can distinguish non-isomorphic graphs and their subsets under suitable parameterizations, making it theoretical expressive for general graph tasks.

## 5 EXPERIMENTS

In this section, we conduct a comprehensive empirical evaluation of ENGNN across graph, node, link, and subgraph-level tasks to demonstrate its broad effectiveness and scalability. As prior work has often focused on only one of these tasks, we use different relevant baselines for each task type.

We evaluate our two primary variants: ENGNN-P (using a permutation-equivariant aggregator, see Appendix G for details) and ENGNN-O (using an orthogonal-transformation-equivariant aggregator). All ENGNN models use noise features sampled i.i.d. from a normal distribution. And the noise is resampled in each forward pass following previous work (Abboud et al., 2021). For ablation studies, we compare against a vanilla MPNN (without noise) and a Noise MPNN (NMPNN), which represents the inequivariant noise approach from prior work. All models are trained in a supervised manner. As our main focus is real-world performance, experiments verifying generalization gain are

Table 1: Roc-auc score ↑ of ENGNN and rGIN on synthetic and TU dataset.

| dataset | TRI(N) | TRI(X) | LCC(N) | LCC(X) | MDS(N) | MDS(X) | MUTAG | NCI1 | PROTEINS |
|---|---|---|---|---|---|---|---|---|---|
| GINs | 0.500 | 0.500 | 0.500 | 0.500 | 0.500 | 0.500 | $0.946_{\pm0.034}$ | $0.870_{\pm0.009}$ | $0.806_{\pm0.029}$ |
| rGINs | 0.908 | 0.926 | 0.811 | 0.852 | 0.807 | 0.810 | $0.949_{\pm0.040}$ | $0.876_{\pm0.010}$ | $0.810_{\pm0.020}$ |
| MPNN | 0.500 | 0.500 | 0.500 | 0.500 | 0.500 | 0.500 | $0.954_{\pm0.007}$ | $0.892_{\pm0.005}$ | $0.831_{\pm0.037}$ |
| NMPNN | 1.000 | 1.000 | 1.000 | 1.000 | 0.933 | 0.932 | $0.972_{\pm0.054}$ | $0.882_{\pm0.008}$ | $0.827_{\pm0.028}$ |
| ENGNN-P | 1.000 | 1.000 | 1.000 | 1.000 | 0.936 | 0.934 | $0.990_{\pm0.019}$ | $0.897_{\pm0.013}$ | $0.837_{\pm0.027}$ |
| ENGNN-O | **1.000** | **1.000** | **1.000** | **1.000** | **0.938** | **0.939** | $\mathbf{0.991_{\pm0.014}}$ | $\mathbf{0.902_{\pm0.022}}$ | $\mathbf{0.843_{\pm0.028}}$ |

Table 2: Mean absolute error on substructures counting. The colored cell means an error ≤ 0.01.

| Method | 3-Cyc. | 4-Cyc. | 5-Cyc. | 6-Cyc. | Tail Tri | Chor Cyc | 4-Cliq. | 4-Path | Tri-Rect |
|---|---|---|---|---|---|---|---|---|---|
| GIN | 0.3515 | 0.2742 | 0.2088 | 0.1555 | 0.3631 | 0.3114 | 0.1645 | 0.1592 | 0.2979 |
| NGNN | 0.0003 | 0.0013 | 0.0402 | 0.0439 | 0.1044 | 0.0392 | 0.0045 | 0.0244 | 0.0729 |
| GNNAK+ | 0.0004 | 0.0041 | 0.0133 | 0.0238 | 0.0043 | 0.0112 | 0.0049 | 0.0075 | 0.1311 |
| PPGN | 0.0003 | 0.0009 | 0.0036 | 0.0071 | 0.0026 | 0.0015 | 0.1646 | 0.0041 | 0.0144 |
| I2GNN | 0.0003 | 0.0016 | 0.0028 | 0.0082 | 0.0011 | 0.0010 | 0.0003 | 0.0041 | 0.0013 |
| DRFWL | 0.0004 | 0.0015 | 0.0034 | 0.0087 | 0.0030 | 0.0026 | 0.0009 | 0.0081 | 0.0070 |
| MPNN | 0.1960 | 0.1808 | 0.1658 | 0.1313 | 0.1585 | 0.1294 | 0.0598 | 0.0594 | 0.1400 |
| NMPNN | 0.0031 | 0.0121 | 0.0167 | 0.0228 | 0.0182 | 0.0179 | 0.0128 | 0.0168 | 0.0572 |
| ENGNN-P | 0.0030 | 0.0047 | 0.0058 | 0.0078 | 0.0038 | 0.0031 | 0.0016 | 0.0033 | 0.0065 |
| ENGNN-O | 0.0031 | 0.0062 | 0.0087 | 0.0092 | 0.0093 | 0.0065 | 0.0023 | 0.0099 | 0.0192 |

in Appendix I. Detailed experimental settings for ENGNN, its ablation variants, and the baselines can be found in Appendix E. Dataset statistics and splits are presented in Appendix F.

**Whole Graph Tasks.** We first evaluate ENGNN on graph-level tasks.

First, we compare our ENGNN with naive noise MPNNs, specifically rGIN (Sato et al., 2021), on six synthetic and three TU datasets (Ivanov et al., 2019), following their original experimental setup. As shown in Table 1, both ENGNN-P and ENGNN-O consistently outperform all baselines across all tasks, showing that **ENGNN significantly outperforms vanilla noise methods**. We also compare our model with other noise methods CLIP (Dasoulas et al., 2020), RP (Murphy et al., 2019), IRNI (Franks et al., 2023), GSPE (Franks et al., 2025; Eliasof et al., 2023), our ENGNN still outperforms these noise methods. The results are shown in Appendix I.

Table 3: Graph property prediction Results.

| | zinc MAE↓ | zinc-full MAE↓ | molhiv AUC↑ |
|---|---|---|---|
| GIN | $0.163_{\pm0.004}$ | $0.088_{\pm0.002}$ | $77.07_{\pm1.49}$ |
| NGNN | $0.111_{\pm0.003}$ | $0.029_{\pm0.001}$ | $78.34_{\pm1.86}$ |
| GNNAK+ | $0.080_{\pm0.001}$ | – | $\mathbf{79.61_{\pm1.19}}$ |
| PPGN | $0.079_{\pm0.005}$ | $0.022_{\pm0.003}$ | - |
| I2GNN | $0.083_{\pm0.001}$ | $0.023_{\pm0.002}$ | $78.68_{\pm0.93}$ |
| DRFWL | $0.077_{\pm0.002}$ | $0.025_{\pm0.003}$ | $78.18_{\pm2.19}$ |
| MPNN | $0.131_{\pm0.007}$ | $0.046_{\pm0.002}$ | $78.27_{\pm1.14}$ |
| NMPNN | $0.136_{\pm0.007}$ | $0.051_{\pm0.004}$ | $77.74_{\pm0.98}$ |
| ENGNN-P | $0.091_{\pm0.005}$ | $0.026_{\pm0.003}$ | $78.51_{\pm0.86}$ |
| ENGNN-O | $\mathbf{0.070_{\pm0.006}}$ | $\mathbf{0.022_{\pm0.003}}$ | $78.63_{\pm0.93}$ |

Next, we benchmark ENGNN against computationally expensive, highly expressive models, including NGNN (Zhang & Li, 2021), GN-NAK+ (Zhao et al., 2022), I2GNN (Huang et al., 2023b), PPGN (Maron et al., 2019a), DR-FWL (Zhou et al., 2023b), as well as the expressive MPNN variant GIN (Xu et al., 2019). We evaluate performance on a subgraph counting task, where the goal is to regress the number of occurrences of various subgraphs, and on three graph property prediction datasets: zinc, zinc-full (Gómez-Bombarelli et al., 2016), and ogbg-molhiv (Hu et al., 2020).

For the subgraph counting task, we follow the setup in Zhou et al. (2023b), where a model is considered capable of counting a subgraph if it achieves a loss lower than 0.01. Results are shown in Table 2. The evaluated subgraphs include 3–6-Cyc (cycles of length 3 to 6), Tail-Tri (Tailed Triangle), Chor-Cyc (cycle with a chord), 4-Cliq (4-Clique), 4-Path (path of length 4), and Tri-Rect (a triangle connected to a rectangle). Vanilla MPNN fails to count any complex subgraphs. In contrast,

Table 4: Results on node classification datasets: Mean accuracy (%) $\pm$ standard variation.

| Dataset | Cora | Citeseer | Pubmed | Computers | Photo | Chameleon | Actor | Squirrel |
|---|---|---|---|---|---|---|---|---|
| GCN | $87.14_{\pm1.01}$ | $79.86_{\pm0.67}$ | $86.74_{\pm0.27}$ | $83.32_{\pm0.33}$ | $88.26_{\pm0.73}$ | $59.61_{\pm2.21}$ | $33.23_{\pm1.16}$ | $46.78_{\pm0.87}$ |
| GIN | $86.58_{\pm0.97}$ | $77.11_{\pm0.76}$ | $86.93_{\pm0.26}$ | $58.87_{\pm7.55}$ | $87.13_{\pm4.52}$ | $66.87_{\pm2.72}$ | $36.66_{\pm7.53}$ | $40.53_{\pm1.16}$ |
| GAT | $88.03_{\pm0.79}$ | $80.52_{\pm0.71}$ | $87.04_{\pm0.24}$ | $83.23_{\pm0.39}$ | $90.94_{\pm0.68}$ | $63.13_{\pm1.93}$ | $33.93_{\pm2.47}$ | $44.49_{\pm0.88}$ |
| APPNP | $88.14_{\pm0.73}$ | $\mathbf{80.47_{\pm0.74}}$ | $88.12_{\pm0.31}$ | $85.32_{\pm0.37}$ | $88.51_{\pm0.31}$ | $51.84_{\pm1.82}$ | $39.66_{\pm0.55}$ | $34.71_{\pm0.57}$ |
| ChebyNet | $86.67_{\pm0.82}$ | $79.11_{\pm0.75}$ | $87.95_{\pm0.28}$ | $87.54_{\pm0.43}$ | $93.77_{\pm0.32}$ | $59.28_{\pm1.25}$ | $37.61_{\pm0.89}$ | $40.55_{\pm0.42}$ |
| GPRGNN | $88.57_{\pm0.69}$ | $80.12_{\pm0.83}$ | $88.46_{\pm0.33}$ | $86.85_{\pm0.25}$ | $93.85_{\pm0.28}$ | $67.28_{\pm1.09}$ | $39.92_{\pm0.67}$ | $50.15_{\pm1.92}$ |
| BernNet | $88.52_{\pm0.95}$ | $80.09_{\pm0.79}$ | $88.48_{\pm0.41}$ | $87.64_{\pm0.44}$ | $93.63_{\pm0.35}$ | $68.29_{\pm1.58}$ | $41.79_{\pm1.01}$ | $51.35_{\pm0.73}$ |
| MPNN | $87.36_{\pm0.52}$ | $79.62_{\pm0.75}$ | $89.53_{\pm0.29}$ | $89.53_{\pm0.83}$ | $94.74_{\pm0.25}$ | $67.18_{\pm1.07}$ | $40.41_{\pm1.53}$ | $51.99_{\pm1.78}$ |
| NMPNN | $20.11_{\pm2.01}$ | $20.80_{\pm2.63}$ | $69.28_{\pm3.14}$ | $66.42_{\pm1.39}$ | $65.12_{\pm1.95}$ | $41.25_{\pm1.38}$ | $23.73_{\pm2.36}$ | $38.25_{\pm1.04}$ |
| ENGNN-P | $88.85_{\pm0.96}$ | $79.97_{\pm0.79}$ | $\mathbf{89.79_{\pm0.64}}$ | $\mathbf{90.48_{\pm0.31}}$ | $\mathbf{95.24_{\pm0.58}}$ | $71.40_{\pm1.29}$ | $40.64_{\pm0.67}$ | $52.77_{\pm1.43}$ |
| ENGNN-O | $\mathbf{89.32_{\pm1.66}}$ | $79.67_{\pm0.70}$ | $89.32_{\pm0.50}$ | $87.96_{\pm0.91}$ | $94.00_{\pm0.80}$ | $\mathbf{71.51_{\pm2.51}}$ | $\mathbf{45.76_{\pm1.85}}$ | $\mathbf{64.66_{\pm1.25}}$ |

Table 5: Results on link prediction benchmarks. OOM means out of GPU memory.

| | Cora | Citeseer | Pubmed | Collab | PPA | DDI |
|---|---|---|---|---|---|---|
| Metric | HR@100 | HR@100 | HR@100 | HR@50 | HR@100 | HR@20 |
| CN | $33.92_{\pm0.46}$ | $29.79_{\pm0.90}$ | $23.13_{\pm0.15}$ | $56.44_{\pm0.00}$ | $27.65_{\pm0.00}$ | $17.73_{\pm0.00}$ |
| AA | $39.85_{\pm1.34}$ | $35.19_{\pm1.33}$ | $27.38_{\pm0.11}$ | $64.35_{\pm0.00}$ | $32.45_{\pm0.00}$ | $18.61_{\pm0.00}$ |
| RA | $41.07_{\pm0.48}$ | $33.56_{\pm0.17}$ | $27.03_{\pm0.35}$ | $64.00_{\pm0.00}$ | $49.33_{\pm0.00}$ | $27.60_{\pm0.00}$ |
| GCN | $66.79_{\pm1.65}$ | $67.08_{\pm2.94}$ | $53.02_{\pm1.39}$ | $44.75_{\pm1.07}$ | $18.67_{\pm1.32}$ | $37.07_{\pm5.07}$ |
| SAGE | $55.02_{\pm4.03}$ | $57.01_{\pm3.74}$ | $39.66_{\pm0.72}$ | $48.10_{\pm0.81}$ | $16.55_{\pm2.40}$ | $53.90_{\pm4.74}$ |
| SEAL | $81.71_{\pm1.30}$ | $83.89_{\pm2.15}$ | $75.54_{\pm1.32}$ | $64.74_{\pm0.43}$ | $48.80_{\pm3.16}$ | $30.56_{\pm3.86}$ |
| NBFnet | $71.65_{\pm2.27}$ | $74.07_{\pm1.75}$ | $58.73_{\pm1.99}$ | OOM | OOM | $4.00_{\pm0.58}$ |
| Neo-GNN | $80.42_{\pm1.31}$ | $84.67_{\pm2.16}$ | $73.93_{\pm1.19}$ | $57.52_{\pm0.37}$ | $\underline{49.13_{\pm0.60}}$ | $63.57_{\pm3.52}$ |
| BUDDY | $\underline{88.00_{\pm0.44}}$ | $\mathbf{92.93_{\pm0.27}}$ | $74.10_{\pm0.78}$ | $\mathbf{65.94_{\pm0.58}}$ | $\mathbf{49.85_{\pm0.20}}$ | $\mathbf{78.51_{\pm1.36}}$ |
| MPNN | $86.26_{\pm1.64}$ | $90.40_{\pm1.71}$ | $79.48_{\pm3.74}$ | $62.84_{\pm1.07}$ | $5.62_{\pm2.52}$ | $24.76_{\pm15.29}$ |
| NMPNN | $48.12_{\pm11.94}$ | $68.63_{\pm7.29}$ | $63.96_{\pm1.92}$ | $7.35_{\pm7.04}$ | $39.90_{\pm5.52}$ | $23.08_{\pm5.89}$ |
| ENGNN-P | $\mathbf{88.10_{\pm1.67}}$ | $91.56_{\pm1.02}$ | $81.26_{\pm1.20}$ | $63.69_{\pm0.82}$ | $44.97_{\pm0.74}$ | $27.64_{\pm6.21}$ |
| ENGNN-O | $87.96_{\pm1.63}$ | $\underline{88.12_{\pm0.97}}$ | $\mathbf{82.08_{\pm2.16}}$ | $\underline{65.34_{\pm0.45}}$ | $48.44_{\pm1.93}$ | $\underline{77.61_{\pm4.50}}$ |

ENGNN-P successfully counts all target subgraphs, and ENGNN-O performs competitively. Notably, while NMPNN fails to meet the success threshold, it still performs far better than the MPNN, confirming that **equivariant use of noise provides a effective expressivity boost**.

The results on graph property prediction datasets are in Table 3. On the zinc and zinc-full datasets, ENGNN-O outperforms all other models, while ENGNN-P also achieves competitive performance. However, on molhiv, ENGNN is outperformed by GNNAK+ and I2GNN, which may be attributed to the inductive bias provided by subgraph-based GNNs. Nevertheless, our ENGNN still achieves strong results on this task. Notably, ENGNN is significantly more scalable than both high-order and subgraph GNNs, requiring as little as 10% of the time and GPU memory compared to subgraph GNNs, as shown in Table 7. **These results demonstrate that ENGNN achieves performance comparable to powerful specialist models at a fraction of the computational cost**

**Node Tasks.** We evaluate our models on real-world node classification tasks. Following previous work (Chien et al., 2021), we use 8 node classification datasets including 5 homogeneous graphs Cora, CiteSeer, PubMed (Yang et al., 2016), Photo, and Amazon (Shchur et al., 2018), and 3 heterogeneous graphs Chameleon, Squirrel (Rozemberczki et al., 2021), and Actor (Pei et al., 2020). Our baselines includes widely used node classification GNNs: GCN (Kipf & Welling, 2016), APPNP (Klicpera et al., 2019), ChebyNet (Defferrard et al., 2016), GPRGNN (Chien et al., 2021), and BernNet (He et al., 2021). The experimental results are presented in Table 4. ENGNN surpasses all baselines on 7/8 datasets, showing its **strong capacity for node tasks**.

**Link Tasks.** We evaluate our models on link prediction datasets, including three citation graphs (Yang et al., 2016) (Cora, Citeseer, and Pubmed) and three Open Graph Benchmark (Hu et al., 2020) datasets (Collab, PPA, and DDI). We employ a range of baseline methods, encompassing tra-

Table 6: Mean Micro-F1 with standard error of the mean on subgraph tasks.

| Method | density | cut ratio | coreness | ppi-bp | hpo-metab | hpo-neuro | em-user |
|---|---|---|---|---|---|---|---|
| GLASS | $0.930_{\pm 0.009}$ | $0.935_{\pm 0.006}$ | $0.840_{\pm 0.009}$ | $\mathbf{0.619}_{\pm 0.007}$ | $\mathbf{0.614}_{\pm 0.005}$ | $\mathbf{0.685}_{\pm 0.005}$ | $0.888_{\pm 0.006}$ |
| SubGNN | $0.919_{\pm 0.006}$ | $0.629_{\pm 0.013}$ | $0.659_{\pm 0.031}$ | $0.599_{\pm 0.008}$ | $0.537_{\pm 0.008}$ | $0.644_{\pm 0.006}$ | $0.816_{\pm 0.013}$ |
| Sub2Vec | $0.459_{\pm 0.012}$ | $0.354_{\pm 0.014}$ | $0.360_{\pm 0.019}$ | $0.388_{\pm 0.001}$ | $0.472_{\pm 0.010}$ | $0.618_{\pm 0.003}$ | $0.779_{\pm 0.013}$ |
| MPNN | $0.321_{\pm 0.023}$ | $0.311_{\pm 0.012}$ | $0.545_{\pm 0.024}$ | $0.547_{\pm 0.009}$ | $0.500_{\pm 0.010}$ | $0.587_{\pm 0.004}$ | $0.641_{\pm 0.017}$ |
| NMPNN | $0.321_{\pm 0.023}$ | $0.311_{\pm 0.012}$ | $0.527_{\pm 0.016}$ | $0.516_{\pm 0.010}$ | $0.460_{\pm 0.010}$ | $0.582_{\pm 0.006}$ | $0.896_{\pm 0.006}$ |
| ENGNN-P | $0.572_{\pm 0.021}$ | $0.744_{\pm 0.050}$ | $0.742_{\pm 0.014}$ | $0.581_{\pm 0.007}$ | $0.540_{\pm 0.008}$ | $0.590_{\pm 0.003}$ | $\mathbf{0.902}_{\pm 0.006}$ |
| ENGNN-O | $\mathbf{0.992}_{\pm 0.003}$ | $\mathbf{0.984}_{\pm 0.007}$ | $\mathbf{0.842}_{\pm 0.026}$ | $0.607_{\pm 0.003}$ | $0.573_{\pm 0.004}$ | $0.579_{\pm 0.006}$ | $0.847_{\pm 0.017}$ |

ditional heuristics like CN (Barabási & Albert, 1999), RA (Zhou et al., 2009), and AA (Adamic & Adar, 2003), as well as GAE models, such as GCN (Kipf & Welling, 2016) and SAGE (Hamilton et al., 2017). Additionally, we consider models involving pairwise representations, including SEAL (Zhang & Chen, 2018) and NBFNet (Zhu et al., 2021b), as well as SF-and-MPNN models like Neo-GNN (Yun et al., 2021) and BUDDY (Chamberlain et al., 2023b). The baseline results are sourced from (Chamberlain et al., 2023b). The experimental results are presented in Table 5. **Our ENGNN achieves best or second best performance on 5/6 datasets.**

**Subgraph Tasks.** We evaluate our models on subgraph classification tasks. Datasets include four synthetic datasets: density, cut ratio, coreness, component, and four real-world subgraph datasets, namely ppi-bp, em-user, hpo-metab, hpo-neuro (Alsentzer et al., 2020). We consider three baselines: SubGNN (Alsentzer et al., 2020) with subgraph-level message passing, Sub2Vec (Adhikari et al., 2018) sampling random walks in subgraphs and encoding them with RNN, GLASS (Wang & Zhang, 2022a) using MPNN with labeling trick. The results are shown in Table 6. **Our ENGNN achieves best performance on 4/8 datasets and second best performance on 3/8 datasets.**

**Scalability.** We present the training time per epoch, GPU memory consumption, and training loss curves in Table 7. Our ENGNN-O achieves comparable resource comsumption to simple MPNN method and takes much less time and memory than high-order GNNs.

Table 7: Time (s) per epoch and GPU memory (GB) consumption on zinc with batch size 128.

| | MPNN | ENGNN-O | SUN | SSWL | PPGN |
|---|---|---|---|---|---|
| Time/s | 2.36 | 4.81 | 20.93 | 45.30 | 20.21 |
| Memory/GB | 0.24 | 0.62 | 3.72 | 3.89 | 20.37 |

**Summary of Experiments.** Across diverse graph, node, link, and subgraph-level benchmarks, EN-GNN demonstrates highly competitive performance, often exceeding that of models specifically designed for a single task type. The comprehensive ablation studies confirm our core hypothesis: ENGNN consistently outperforms vanilla MPNNs and naive Noise MPNNs, highlighting the potential of equivariant use of noise.

# 6 CONCLUSION

To bridge the gap between real-world applicability and theoretical universal expressivity, we propose equivariant noise GNN. It that utilize noise equivariantly for better generalization bound. Our approach demonstrates universal theoretical expressivity and excels in real-world performance. It extends the design space of GNN and provides a principled way to utilize noise feature.

# 7 LIMITATIONS

Although the ENGNN introduced in our work shares the same time complexity as traditional MPNNs, the inclusion of noise features introduces additional computational overhead. Furthermore, despite the fact that noise is not task-specific, our approach requires modifications to the aggregator, which means it cannot be seamlessly integrated with other existing GNNs. Future work will focus on further reducing computational complexity and developing a plug-and-play method.

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

## A  THE USE OF LARGE LANGUAGE MODELS (LLMS)

We use Gemini 2.5 flash to proofread our writing in our paper. We also use it to proofread our proof and keep notations consistent.

## B  PROOFS FOR SECTION 4.1

PAC (Probably Approximately Correct) learning theory is widely used for analyzing generalization. Previous works have analyse the generalization of GNNs (Morris et al., 2023; Pellizzoni et al., 2024; Garg et al., 2020), but there results cannot directly apply to our case. Previous works (Alberto et al., 2021; Behrooz & Jegelka, 2023) using PAC to analyze that symmetry can improve the generalization of kernel regression. Elesedy (2022); Mroueh et al. (2015); Sannai et al. (2021); Zhu et al. (2021a) show that symmetry can improve generalization for neural network models. However, they primarily focus on a general learning setting. In this work, we propose to use noise invariance to boost the generalization for GNNs with random features. Some previous works (Alberto et al., 2021; Behrooz & Jegelka, 2023) also introduce noise as auxiliary features into equivariant networks for performing approximately equivariant tasks (that are not strictly equivariant). However, they are equivariant to operations on the original data, not to noise transformations. In this section, we first restate our notations, then give formal definitions for important concepts in Section 4.1, and provide proofs for conclusions in Section 4.1.

**Notations:** Consider a graph $G = (A, X) \in \mathcal{G}$ with $n$ nodes, where $A \in \mathbb{R}^{n \times n}$ is the adjacency matrix and $X \in \mathbb{R}^{n \times d}$ is the node feature matrix. Additionally, there is auxiliary feature noise $Z \in \mathcal{Z} = \mathbb{R}^{n \times C}$. The input domain is $\mathcal{I} = G \times \mathcal{Z}$, and the target domain is $\mathcal{Y} \subseteq \mathbb{R}^{d'}$. We introduce a transformation set $T$. For an operation $t \in T$ on the noise, the group acts on the data $(A, X, Z)$ as $(A, X, t(Z))$.

A learning task is defined as $(I, Y, l)$, where $I$ and $Y$ are random elements in $\mathcal{I}$ and $\mathcal{Y}$, respectively, and $l : \mathcal{Y} \times \mathcal{Y} \to \mathbb{R}^+$ is an integrable, bounded, and $C_l$-Lipschitz loss function. Let $\mathcal{H}$ be a class of measurable functions $h : \mathcal{I} \to \mathcal{Y}$, known as the hypothesis class. An algorithm alg : $\bigcup_{i \in \mathbb{N}} (\mathcal{I} \times \mathcal{Y})^i \to \mathcal{H}$ maps a finite sequence of data points to a hypothesis in $\mathcal{H}$. In our work, we assume the algorithm is $T$-invariant. For example, when $\mathcal{H}$ consists of invariant functions, empirical risk minimization is $T$-invariant.

**Definitions:**

**Definition B.1.** *(Sample Complexity) Algorithm alg learns $\mathcal{H}$ with respect to a task $(I, Y, l)$ if there exists a function $m : (0, 1)^2 \to \mathbb{N}$ such that for all $\epsilon, \delta \in (0, 1)$, if $n > m(\epsilon, \delta)$, then:*

$$P\left(\mathbb{E}\left[l(h_S(I), Y) \mid S\right] \geq \inf_{h \in \mathcal{H}} \mathbb{E}\left[l(h(I), Y)\right] + \epsilon\right) \leq \delta, \tag{10}$$

*where $h_S = alg(S)$ and $S \sim (I, Y)^n$ is an i.i.d. sample. The sample complexity of alg is the minimal $m(\epsilon, \delta)$ satisfying this condition.*

**Definition B.2.** *(Semi-Metric) Given a set $\mathcal{X}$ and a function $\rho : \mathcal{X} \times \mathcal{X} \to \mathbb{R}$, if for all $x, y, z \in \mathcal{X}$:*

- $\rho(x, y) \geq 0$,
- $\rho(x, x) = 0$,
- $\rho(x, y) = \rho(y, x)$,

*then $\rho$ is a semi-metric on $\mathcal{X}$, and $(\mathcal{X}, \rho)$ is a semi-metric space. If $\rho(x, y) > 0$ for all $x \neq y$, $\rho$ is a metric.*

For $G$, $\mathcal{Z}$, and $\mathcal{Y}$, we define semi-metrics $\rho_G$, $\rho_Z$, and $\rho_Y$ as follows:

- $\rho_G$: For graphs $G_1 = (A_1, X_1)$ and $G_2 = (A_2, X_2)$,

$$\rho_G(G_1, G_2) = \left(\|X_1 - X_2\| + \|A_1 - A_2\|\right). \tag{11}$$

  where $S_n$ is the symmetric group.

- $\rho_Z$ and $\rho_Y$: $\ell_1$-norm differences:

$$\rho_Z(Z_1, Z_2) = \|Z_1 - Z_2\|_1, \quad \rho_Y(y_1, y_2) = \|y_1 - y_2\|_1. \tag{12}$$

**Definition B.3.** *(Cover) A $\delta$-cover of a semi-metric space $(\mathcal{X}, \rho)$ is a set $S \subseteq \mathcal{X}$ such that $\forall x \in \mathcal{X}, \exists s \in S, \rho(s, x) \leq \delta$. The covering number $N(\mathcal{X}, \rho, r)$ is the minimal $|S|$ for radius $r$.*

### B.1 PROOF FOR THEOREM 4.1

We first bound the sample complexity using the covering number of the hypothesis space.

**Lemma B.4.** *The sample complexity for an algorithm with $C_l$-Lipschitz loss function is*

$$O\left(\frac{1}{\epsilon^2}\left(\ln N\left(\mathcal{H}, \rho_H, \frac{\epsilon}{4C_l}\right) + \ln \frac{1}{\delta}\right)\right), \tag{13}$$

*where $\rho_H$ is a semi-metric on $\mathcal{H}$, defined as $\rho_H(h, h') = \sup_{I \in \mathcal{I}} \|h(I) - h'(I)\|$.*

*Proof.* Given random variables $I, Y$, define

$$L(h) = \mathbb{E}[l(h(I), Y)] - \frac{1}{n}\sum_{i=1}^{n} l(h(I_i), Y_i). \tag{14}$$

For $h, h' \in \mathcal{H}$:

$$|L(h) - L(h')| \leq \mathbb{E}\left[|l(h(I), Y) - l(h'(I), Y)|\right] + \frac{1}{n}\sum_{i=1}^{n} |l(h(I_i), Y_i) - l(h'(I_i), Y_i)| \tag{15}$$

$$\leq 2C_l \rho_H(h, h'). \tag{16}$$

Let $K$ be a $\kappa$-cover of $\mathcal{H}$, and define $D(k) = \{h \in \mathcal{H} \mid \rho_H(h, k) \leq \kappa\}$. Then:

$$\mathbb{P}\left[\sup_{h \in \mathcal{H}} |L(h)| \geq \epsilon\right] \leq \sum_{k \in K} \mathbb{P}\left[\sup_{h \in D(k)} |L(h)| \geq \epsilon\right] \tag{17}$$

$$\leq \sum_{k \in K} \mathbb{P}\left[|L(k)| + 2C_l\kappa \geq \epsilon\right] \tag{18}$$

$$\leq \sum_{k \in K} \mathbb{P}\left[|L(k)| \geq (1 - \alpha)\epsilon\right], \quad \alpha = \frac{2C_l\kappa}{\epsilon}. \tag{19}$$

By Hoeffding's inequality (assuming $l(h(I), Y) \in [0, 1]$):

$$\mathbb{P}\left[|L(k)| \geq (1 - \alpha)\epsilon\right] \leq 2 \exp\left(-2n(1 - \alpha)^2 \epsilon^2\right). \tag{20}$$

For $\alpha = \frac{1}{2}$:

$$\mathbb{P}\left[\sup_{h \in \mathcal{H}} |L(h)| \geq \epsilon\right] \leq 2N\left(\mathcal{H}, \rho_H, \frac{\epsilon}{4C_l}\right) \exp\left(-\frac{n\epsilon^2}{2}\right). \tag{21}$$

Thus, the sample complexity is $O\left(\frac{1}{\epsilon^2}\left(\ln N\left(\mathcal{H}, \rho_H, \frac{\epsilon}{4C_l}\right) + \ln \frac{1}{\delta}\right)\right)$. □

We further decompose the input space $\mathcal{I}$ into $G \times \mathcal{Z}$:

**Lemma B.5.** *If $\mathcal{H}$ contains functions partially Lipschitz in $\mathcal{G}$ (constant $C_G$) and $\mathcal{Z}$ (constant $C_Z$):*

$$N(\mathcal{H}, \rho_H, r) \leq N\left(\mathcal{Y}, \rho_Y, \frac{r}{3}\right)^{N\left(G, \rho_G, \frac{r}{3C_G}\right)N\left(\mathcal{Z}, \rho_{Z,T}, \frac{r}{3C_Z}\right)}. \tag{22}$$

*Proof.* Let $I$, $J$, and $K$ be $r_1$-, $r_2$-, and $r_3$-covers of $G$, $\mathcal{Z}$, and $\mathcal{Y}$, respectively. Construct:

$$F = \left\{ f_k \mid k \in K^{|I| \times |J|}, f_k(G, Z) = k_{ij} \text{ if } G \in D(I_i), Z \in D(J_j) \right\}, \tag{23}$$

where $D(I_i)$ is intuitively the set of graph close to $I_i$: $D(I_i) \subseteq \mathcal{G}, D(I_i) \cap D(I_j) = $ if $i \neq j$, $\forall G \in D(I_i), \rho_G(G, I_i) \leq r_1$, and $\cup_i D(I_i) = \mathcal{G}$. $D(J_j)$ is defined for $\mathcal{Z}$ similarly.

For all $h \in \mathcal{H}$:

$$\min_{f \in F} \rho_H(f, h) \leq \min_{f \in F} \max_{i \in I} \max_{j \in J} \sup_{G \in D(I_i)} \sup_{Z \in D(J_j)} \left(\|f(G, Z) - f(I_i, Z)\|\right. \tag{24}$$

$$+ \|f(I_i, Z) - f(I_i, J_j)\| + \|f(I_i, J_j) - h(I_i, J_j)\|) \tag{25}$$

$$\leq C_G r_1 + C_Z r_2 + r_3. \tag{26}$$

Setting $r_1 = \frac{r}{3C_G}$, $r_2 = \frac{r}{3C_Z}$, and $r_3 = \frac{r}{3}$, we obtain:

$$N(\mathcal{H}, \rho_H, r) \leq N\left(\mathcal{Y}, \rho_Y, \frac{r}{3}\right)^{N\left(G, \rho_G, \frac{r}{3C_G}\right)N\left(\mathcal{Z}, \rho_{Z,T}, \frac{r}{3C_Z}\right)}. \tag{27}$$

□

The sample complexity becomes:

$$O\left(\frac{1}{\epsilon^2}\left(N\left(G, \rho_G, \frac{\epsilon}{12C_l C_G}\right) N\left(\mathcal{Z}, \rho_{Z,T}, \frac{\epsilon}{12C_l C_Z}\right) \ln N\left(\mathcal{Y}, \rho_Y, \frac{\epsilon}{12C_l}\right) + \ln \frac{1}{r}\right)\right). \tag{28}$$

## B.2 PROOF FOR PROPOSITION 4.2

**Proposition B.6.** *If $T_1 \subseteq T_2$, then for all $r > 0$, $N(\mathcal{Z}, \rho_{Z,T_1}, r) \geq N(\mathcal{Z}, \rho_{Z,T_2}, r)$.*

*Proof.* For all $Z_1, Z_2 \in \mathcal{Z}$,

$$\rho_{Z,T_1}(Z_1, Z_2) = \inf_{t,t' \in T_1} \rho_Z(t(Z_1), t'(Z_2)) \geq \inf_{t,t' \in T_2} \rho_Z(t(Z_1), t'(Z_2)) = \rho_{Z,T_2}(Z_1, Z_2). \tag{29}$$

Let $S$ be an $r$-cover of $(\mathcal{Z}, \rho_{Z,T_1})$ with $|S| = N(\mathcal{Z}, \rho_{Z,T_1}, r)$. For any $Z \in \mathcal{Z}$, there exists $Z' \in S$ such that $\rho_{Z,T_2}(Z, Z') \leq \rho_{Z,T_1}(Z, Z') \leq r$. Thus, $S$ is also an $r$-cover for $(\mathcal{Z}, \rho_{Z,T_2})$, implying $N(\mathcal{Z}, \rho_{Z,T_2}, r) \leq N(\mathcal{Z}, \rho_{Z,T_1}, r)$. □

### B.3 PROOF FOR PROPOSITION 4.3

**Proposition B.7.** *If $\mathcal{Z} = [0,1]^{n \times C}$ and $T$ includes all permutations of $C$ channels, then for sufficiently small $r$:*

$$\frac{N(\mathcal{Z}, \rho_{Z,T}, r)}{N(\mathcal{Z}, \rho_Z, r)} \leq \frac{1}{C!}. \tag{30}$$

*Proof.* For $\rho_Z$-covers under the $\ell_1$-metric, consider the grid:

$$S = \left\{ 2r \cdot \mathbf{k} + r \mid \mathbf{k} \in \{0, 1, \ldots, \lceil 1/(2r) \rceil\}^{n \times C} \right\}. \tag{31}$$

The covering number satisfies $\left(\frac{1}{2r}\right)^{nC} \leq N(\mathcal{Z}, \rho_Z, r) \leq \lceil 1/(2r) \rceil^{nC}$, as the volume of $2r$-cubes covers $[0,1]^{nC}$.

Define subsets of $S$:

$$S' = \{z \in S \mid \forall 0 \leq i < j < C, z_{0,i} \neq z_{0,j}\},$$
$$S'' = \{z \in S \mid \forall 0 \leq i < j < C, z_{0,i} < z_{0,j}\}.$$

Permuting channels of $S''$ generates $S'$, so $|S'| = C! \cdot |S''|$. The set $(S \setminus S') \cup S''$ forms an $r$-cover for $(\mathcal{Z}, \rho_{Z,T})$.

As $r \to 0^+$, $|S \setminus S'| = O\left((1/r)^{nC-1}\right)$ is negligible compared to $N(\mathcal{Z}, \rho_Z, r) = \Theta\left((1/r)^{nC}\right)$. Thus,

$$\lim_{r \to 0^+} \frac{N(\mathcal{Z}, \rho_{Z,T}, r)}{N(\mathcal{Z}, \rho_Z, r)} = \frac{|S''|}{|S|} = \frac{1}{C!}. \tag{32}$$

$\square$

## C PROBABILITY FOR DISTRIBUTION COLLISION

For continuous distributions, the probability that two independently sampled noise vectors coincide is 0. However, as neural networks are typically continuous, they may still produce similar outputs for similar noise inputs. We define the coincidence between two noise vectors $\mathbf{z}_1, \mathbf{z}_2$ as $\|\mathbf{z}_1 - \mathbf{z}_2\|_1 \leq \delta$, where $\delta \in \mathbb{R}^+$.

**Proposition C.1.** *Given $n$ vectors $\mathbf{z}_1, \mathbf{z}_2, \ldots, \mathbf{z}_n$ independently sampled from a uniform distribution over $[0,1]^C$, the probability that $\forall i \neq j, \|\mathbf{z}_i - \mathbf{z}_j\|_1 \geq \delta$ is bounded by:*

$$\prod_{i=1}^{n-1} \left[ 1 - i \left(\frac{\delta}{2}\right)^C \right] \quad \text{if } n \leq \left(\frac{\delta}{2}\right)^{-C} + 1, \tag{33}$$

*and $0$ otherwise.*

*Proof.* For each $\mathbf{z}_i$, define a hypercube:

$$D(\mathbf{z}_i) = \left\{ \mathbf{z} \in [0,1]^C \mid \|\mathbf{z} - \mathbf{z}_i\|_1 \leq \frac{\delta}{2} \right\}. \tag{34}$$

To ensure $\|\mathbf{z}_i - \mathbf{z}_j\|_1 \geq \delta$ for all $i \neq j$, the hypercubes $D(\mathbf{z}_i)$ and $D(\mathbf{z}_j)$ must be disjoint. Let $p_m$ denote the probability of placing $m$ non-overlapping hypercubes. The recursion is:

$$p_{m+1} = p_m \left( 1 - m \left(\frac{\delta}{2}\right)^C \right), \tag{35}$$

since each existing hypercube occupies at least $\left(\frac{\delta}{2}\right)^C$ volume in $[0,1]^C$. Solving recursively gives the product bound. For $n > \left(\frac{\delta}{2}\right)^{-C} + 1$, the probability vanishes as the total volume of hypercubes exceeds 1. $\square$

When $\delta < 1$, larger $C$ reduces the collision probability due to the exponential decay of $\left(\frac{\delta}{2}\right)^C$.

# D    PROOF FOR SECTION 4

## D.1    PROOF FOR THEOREM 4.4

Since the aggregator produce one single equivariant representation with a set of equivariant representations as input, the transformation must be row-wise on noise matrix.

**Theorem D.1.** *(Invariance) If the aggregator is equivariant to a set of noise transformations $T$, then ENGNN's outputs for graphs and subsets are invariant to $T$.*

*Proof.* Suppose the input noise $Z^{(0)}$ is transformed with row-wise transformation $t$. Let $X_i, Z_i, h_G, h_U$ denote original representations for node $i$, graph, and subset $U$. Let $\hat{X}_i, \hat{Z}_i, \hat{h}_G, \hat{h}_U$ denote the output for transformed input. We are going to prove that $\hat{h}_G = h_G$ and $\hat{h}_U = h_U$.

**Input Layer:** For $k = 0$, the transformed noise is row-wise:

$$\hat{Z}_i^{(0)} = t\left(Z_i^{(0)}\right), \quad \hat{X}_i^{(0)} = X_i^{(0)}. \tag{36}$$

**Equivariance of Message-Passing:** Assume at layer $k$, the node features and noise satisfy:

$$\hat{X}_i^{(k)} = X_i^{(k)}, \quad \hat{Z}_i^{(k)} = t\left(Z_i^{(k)}\right). \tag{37}$$

At layer $k + 1$, the update rule preserves equivariance:

$$\left(\hat{X}_i^{(k+1)}, \hat{Z}_i^{(k+1)}\right) = \text{AGGR}_1\left[\left\{\text{AGGR}_2\left(\left\{\left(\hat{X}_j^{(k)}, \hat{Z}_j^{(k)}\right) \mid j \in \mathcal{N}(i)\right\}\right), \right.\right. \tag{38}$$

$$\left.\left.\left(\text{MLP}\left(\hat{X}_i^{(k)}\right), \hat{Z}_i^{(k)}\right)\right\}\right] \tag{39}$$

$$= \text{AGGR}_1\left[\left\{\text{AGGR}_2\left(\left\{\left(X_j^{(k)}, t\left(Z_j^{(k)}\right)\right) \mid j \in \mathcal{N}(i)\right\}\right), \right.\right. \tag{40}$$

$$\left.\left.\left(\text{MLP}\left(X_i^{(k)}\right), t\left(Z_i^{(k)}\right)\right)\right\}\right] \tag{41}$$

$$= \left(X_i^{(k+1)}, t\left(Z_i^{(k+1)}\right)\right). \tag{42}$$

By induction, message-passing layers preserve equivariance.

**Invariance of Graph Representation:**

The graph representation aggregates equivariant node features:

$$\left(\hat{h}_G, \hat{Z}_G\right) = \text{AGGR}\left(\left\{\left(\hat{X}_i, \hat{Z}_i\right) \mid i \in V\right\}\right) \tag{43}$$

$$= \text{AGGR}\left(\left\{(X_i, t(Z_i)) \mid i \in V\right\}\right) \tag{44}$$

$$= (h_G, t(Z_G)). \tag{45}$$

Since AGGR is invariant under row-wise transformations, $\hat{h}_G = h_G$.

**Invariance of Subset Representation:**

$$\left(\hat{h}'_U, \hat{Z}'_U\right) = \text{AGGR}_1\left(\left\{\left(\hat{X}_i, \hat{Z}_i\right) \mid i \in U\right\}\right) \tag{46}$$

$$= \text{AGGR}_1\left(\left\{(X_i, t(Z_i)) \mid i \in U\right\}\right) \tag{47}$$

$$= (h'_U, t(Z'_U)). \tag{48}$$

**Final Subset Representation:**

$$\left(\hat{h}_U, \hat{Z}_U\right) = \text{AGGR}_2\left[\left\{\left(\text{MLP}\left(\hat{h}_G\right), \hat{Z}_G\right), \left(\hat{h}'_U, \hat{Z}'_U\right)\right\}\right] \tag{49}$$

$$= \text{AGGR}_2\left[\left\{(\text{MLP}(h_G), t(Z_G)), (h'_U, t(Z'_U))\right\}\right] \tag{50}$$

$$= (h_U, t(Z_U)). \tag{51}$$

By the equivariance of $\text{AGGR}_2$, $\hat{h}_U = h_U$.    $\square$

## D.2 PROOF FOR THEOREM 4.5

**Theorem D.2.** *Assume each node has distinct random features, and the aggregator in ENGNN achieves universal expressivity (e.g. Aggregators in Appendix G).*

*(Graph-Level Expressivity) Let ENGNN$(G, Z)$ denote the model's output for graph $G$ and noise $Z$. For any non-isomorphic graphs $G$ and $H$, there exists a parameterization such that:*

$$ENGNN(G, Z_1) \neq ENGNN(H, Z_2), \quad \forall Z_1, Z_2 \in \mathcal{Z}. \tag{52}$$

*(Subgraph-Level Expressivity) Let ENGNN$(U, G, Z)$ denote the output for subset $U$ in graph $G$. For any non-isomorphic pairs $(G, U_G)$ and $(H, U_H)$, there exists a parameterization such that:*

$$ENGNN(U_G, G, Z_1) \neq ENGNN(U_H, H, Z_2), \quad \forall Z_1, Z_2 \in \mathcal{Z}. \tag{53}$$

*Proof.* **Graph-Level Expressivity** Suppose that there exist two non-isomorphic graphs $G$ and $H$, distinct node noise features $Z \in \mathcal{Z}$ that for all $i \neq j, Z_i \neq Z_j$. For non-isomorphic $G$ and $H$, the universal aggregator constructs injective mappings:

Let $u_i, v_i$ denote the invariant and equivariant representations of node $i$, encoding:

$$(u_i, v_i) \xrightarrow{\text{inj}} \left( X_i, Z_i, \{\{(X_j, Z_j) \mid (i,j) \in E\}\} \right). \tag{54}$$

Pooling across nodes yields:

$$\text{AGGR}\left( \{\{(u_i, v_i) \mid i \in V\}\} \right) \xrightarrow{\text{inj}} \left( \{\{(Z_i, Z_j) \mid (i,j) \in E\}\}, \{\{(X_i, Z_i) \mid i \in V\}\} \right). \tag{55}$$

Since $Z_i$ are unique and non-isomorphic graphs have distinct edge sets or node features, the aggregated representation differs.

**Subgraph-Level Expressivity** Suppose that there exist non-isomorphic pairs $(G, U_G)$ and $(H, U_H)$ (no permutation maps $G \to H$ and $U_G \to U_H$).

Node representations $u_i, v_i$ encode:

$$(u_i, v_i) \xrightarrow{\text{inj}} \left( X_i, Z_i, \{\{(X_j, Z_j) \mid (i,j) \in E\}\} \right). \tag{56}$$

Subset aggregation preserves injectivity:

$$\text{AGGR}\left( \{\{(u_i, v_i) \mid i \in U_G\}\} \right) \xrightarrow{\text{inj}} \{\{Z_i \mid i \in U_G\}\}. \tag{57}$$

The full representation combines graph and subset information:

$$\text{ENGNN}(G, U_G, Z) \xrightarrow{\text{inj}} \left( \{\{Z_i \mid i \in U_G\}\}, \{\{(Z_i, Z_j) \mid (i,j) \in E\}\}, \{\{(X_i, Z_i) \mid i \in V\}\} \right). \tag{58}$$

Non-isomorphic pairs $(G, U_G)$ and $(H, U_H)$ yield distinct representations due to unique $Z_i$ and structural differences. $\square$

## E EXPERIMENTAL SETTING

Our code is available at https://anonymous.4open.science/r/EquivNoiseGNN. We use PyTorch and PyTorch Geometric for model development. All experiments are conducted on an Nvidia 4090 GPU on a Linux server. We use the AdamW optimizer with a cosine annealing scheduler. We use L1 loss for regression tasks and cross-entropy loss for classification tasks.

We perform random search using Optuna to optimize hyperparameters by maximize valid score. The selected hyperparameters for each model are available in our code. The hyperparameter we tune include number of layer in [2, 10], hidden dimention in [16, 128], number of noise channel in [8, 64], number of noise feature dimension in [16, 64], learning rate in [1e-4, 1e-2], weight decay in [1e-6, 1e-1]. The detailed hyperparameters are in our code. For baselines, we directly use the score reported in the original paper. For ablation model MPNN, NMPNN, we use the same hyperparameter and model architecture as ENGNN, but use aggregators in Appendix G.3. Our experiment on ZINC, ZINC-FULL, and SubgraphCount takes 5 hours per run, and takes less than 1 hour for other tasks per run. All experiments takes about 200 hours.

Table 8: Statistics of the datasets. #Nodes and #Edges denote the number of nodes and edges per graph. In "Task" column, $k$-CLS means classification with $k$ classes, and REG means regression. In "Split" column, "fixed" means the dataset uses the split provided in the original release, and 10-fold means 10-fold cross validation. Otherwise, it is of the formal training set ratio/valid ratio/test ratio.

| Graph | Name | #Graphs | #Nodes | #Edges | Task | Metric | Split |
|---|---|---|---|---|---|---|---|
| | LCC/TRI/MDS | 3,000 | 20.0 | 30.0 | Node 2-CLS | AUROC | fixed |
| | MUTAG | 188 | 17.9 | 37.6 | 2-CLS | AUROC | 10-fold |
| | NCI1 | 4,110 | 29.9 | 64.6 | 2CLS | AUROC | 10-fold |
| | PROTEINS | 1,113 | 39.1 | 145.6 | 2CLS | AUROC | 10-fold |
| | SubgraphCount | 5,000 | 18.8 | 31.3 | Node REG | MAE | 0.3/0.2/0.5. |
| | ZINC | 12,000 | 23.2 | 24.9 | REG | MAE | fixed |
| | ZINC-full | 249,456 | 23.2 | 24.9 | REG | MAE | fixed |
| | ogbg-molhiv | 41,127 | 25.5 | 27.5 | REG | AUC | fixed |
| Subgraph | Name | #Subgraphs | #Nodes | #Edges | Task | Metric | Split |
| | density | 250 | 5,000 | 29,521 | 3-CLS | F1-score | 0.5/0.25/0.25 |
| | cut-ratio | 250 | 5,000 | 83,969 | 3-CLS | F1-score | 0.5/0.25/0.25 |
| | coreness | 221 | 5,000 | 118,785 | 3-CLS | F1-score | 0.5/0.25/0.25 |
| | ppi_bp | 1,591 | 17,080 | 316,951 | 6-CLS | F1-score | fixed |
| | hpo_metab | 2,400 | 14,587 | 3,238,174 | 6-CLS | F1-score | fixed |
| | hpo_neuro | 4,000 | 14,587 | 3,238,174 | 2-CLS | F1-score | fixed |
| | em_user | 324 | 57,333 | 4,573,417 | 2-CLS | F1-score | fixed |
| Node | Name | - | #Nodes | #Edges | Task | Metric | Split |
| | Cora | | 2,708 | 5,278 | 7-CLS | ACC | 0.6/0.2/0.2 |
| | CiteSeer | | 3,327 | 4,552 | 6-CLS | ACC | 0.6/0.2/0.2 |
| | PubMed | | 19,717 | 44,324 | 3-CLS | ACC | 0.6/0.2/0.2 |
| | Computers | | 13,752 | 245,861 | 10-CLS | ACC | 0.6/0.2/0.2 |
| | Photo | | 7,650 | 119,081 | 8-CLS | ACC | 0.6/0.2/0.2 |
| | Chameleon | | 2,277 | 31,371 | 5-CLS | ACC | 0.6/0.2/0.2 |
| | Squirrel | | 5,201 | 198,353 | 5-CLS | ACC | 0.6/0.2/0.2 |
| | Actor | | 7600 | 26659 | 5-CLS | ACC | 0.6/0.2/0.2 |
| Link | Name | - | #Nodes | #Edges | Task | Metric | Split |
| | Cora | | 2,708 | 5,278 | 2-CLS | Hit@100 | 0.7/0.1/0.2 |
| | Citeseer | | 3,327 | 4,676 | 2-CLS | Hit@100 | 0.7/0.1/0.2 |
| | Pubmed | | 18,717 | 44,327 | 2-CLS | Hit@100 | 0.7/0.1/0.2 |
| | Collab | | 235,868 | 1,285,465 | 2-CLS | Hit@50 | fixed |
| | PPA | | 576,289 | 30,326,273 | 2-CLS | Hit@100 | fixed |
| | DDI | | 4,267 | 1,334,889 | 2-CLS | Hit@20 | fixed |
| | Citation2 | | 2,927,963 | 30,561,187 | 2-CLS | mrr | fixed |

# F  DATASET

We summarize the statistics of all our datasets in Table 8. For graph datasets, SubgraphCount is the dataset used in substructure counting tasks provided by Huang et al. (2023b), they are random graph with the count of substructure as node label. ZINC, ZINC-FULL (Gómez-Bombarelli et al., 2016), and ogbg-molhiv are three datasets of molecules. LCC, TRI, MDS, and SubgraphCount are all node tasks, but previous GNNs for graph tasks also use them to evaluate expressivity, so we consider them as graph tasks. Ogbg-molhiv is one of Open Graph Benchmark dataset, which aims to use graph structure to predict whether a molecule can inhibits HIV virus replication. For subgraph datasets, we use the code provided by SubGNN to produce synthetic datasets and use the real-world datasets provided by SubGNN (Alsentzer et al., 2020) directly. For link prediction datasets, random splits use 70%/10%/20% edges for training/validation/test set respectively. Different from others, the collab dataset allows using validation edges as input on test set. For node classification datasets, random splits use 60%/20%/20% nodes for training/validation/test set respectively.

## G  Equivariant Aggregator

The equivariant aggregator processes a multiset of invariant-equivariant feature pairs $\{(X_i, Z_i) \mid X_i \in \mathbb{R}^d, Z_i \in \mathbb{R}^{L \times C}, i = 1, \ldots, B\}$ to produce an output pair $(X', Z')$. Below, we formalize two variants: $O(C)$(orthogonal transformations on $C$-dimensional noise vectors)- and $S(C)$(permutation on $C$-dimensional noise vectors)-equivariant aggregators, achieving equivariance, universal expressivity, and linear complexity. Prior work (Blum-Smith et al., 2024; Villar et al., 2021; Maron et al., 2020) informs our designs.

### G.1  $O(C)$-Equivariant Aggregator

The aggregator consists of the following steps:

1. Compute an equivariant orientation matrix $C \in \mathbb{R}^{L' \times C}$ via:

$$C = \sum_{i=1}^{B} f_1(Z_i Z_i^T) Z_i, \quad f_1 : \mathbb{R}^{L \times L} \to \mathbb{R}^{L' \times L}. \tag{59}$$

Projections $C Z_i^T$ yield invariant features.

2. Aggregating Invariants: Aggregate invariant features using a DeepSet (Zaheer et al., 2017) $f_2$:

$$X' = f_2\left(\left\{(X_i, C Z_i^T, Z_i Z_i^T, C C^T) \mid i = 1, \ldots, B\right\}\right). \tag{60}$$

3. Scale and Aggregating Equivariant Features: Generate equivariant outputs via MLP $f_3 : \mathbb{R}^{d+d+LL'+L'^2} \to \mathbb{R}^L$:

$$Z' = \sum_{i=1}^{B} f_3\left(X_i, X', C Z_i^T, Z_i Z_i^T, C C^T\right) Z_i. \tag{61}$$

First, we show that this aggregator, denoted as AGGR, is $O(C)$-equivariant.

**Proposition G.1.** *For all $t \in O(C)$, if $\hat{X}', \hat{Z}' = AGGR(\{(X_i, t(Z_i)) \mid i = 1, \ldots, B\})$ and $X', Z' = AGGR(\{(X_i, Z_i) \mid i = 1, \ldots, B\})$, then $\hat{X}' = X'$ and $\hat{Z}' = t(Z')$.*

*Proof.* We analyze each step under orthogonal transformation $t$:

1. Orientation: Since $t$ is orthogonal, $t(Z_i Z_i^T) = t Z_i Z_i^T t^T$. The function $f_1$ operates on $Z_i Z_i^T$ and outputs a matrix in $\mathbb{R}^{L' \times L}$. Applying $t$ to $Z_i$ transforms $C$ as:

$$\hat{C} = \sum_{i=1}^{B} f_1(t Z_i Z_i^T t^T) t Z_i = t\left(\sum_{i=1}^{B} f_1(Z_i Z_i^T) Z_i\right) = tC. \tag{62}$$

2. Invariant Aggregation: The input to $f_2$ includes terms like $C Z_i^T$ and $C C^T$. Under transformation, these become:

$$t C Z_i^T t^T, \quad t C C^T t^T. \tag{63}$$

Since $f_2$ is permutation-invariant and operates on multiset inputs, the aggregate $X'$ remains unchanged ($\hat{X}' = X'$).

3. Equivariant Output: The final step applies $t$ to $Z_i$ and computes:

$$\hat{Z}' = \sum_{i=1}^{B} f_3(\cdots) t Z_i = t\left(\sum_{i=1}^{B} f_3(\cdots) Z_i\right) = tZ', \tag{64}$$

where $\cdots = X_i, X', C Z_i^T, Z_i Z_i^T, C C^T$. Thus, the aggregator is $O(C)$-equivariant. $\square$

Second, we show its universal expressivity.

**Proposition G.2.** *There exists a measure zero set $B \subseteq \mathbb{R}^{n \times d} \times \mathbb{R}^{n \times L \times C}$ such that for all $(X, Z), (\hat{X}, \hat{Z}) \in \mathbb{R}^{n \times d} \times \mathbb{R}^{n \times L \times C} \setminus B$, if $\forall P \in S_n, t \in O(C), (PX, Pt(Z)) \neq (\hat{X}, \hat{Z})$, then $AGGR(X, Z) \neq AGGR(\hat{X}, \hat{Z})$.*

*Proof.* Let $B$ exclude inputs where any two $Z_i$ have equal norms ($\|Z_i\|_2 = \|Z_j\|_2$ for $i \neq j$). For $(X, Z) \notin B$, without loss of generality, we assume sorted norms $\|Z_1\| < \|Z_2\| < \cdots < \|Z_B\|$.

Suppose $\text{AGGR}(X, Z) = \text{AGGR}(\hat{X}, \hat{Z})$. From the invariant aggregation step:

$$\{(X_i, CZ_i^T, Z_i Z_i^T, CC^T)\} = \{(\hat{X}_i, \hat{C}\hat{Z}_i^T, \hat{Z}_i \hat{Z}_i^T, \hat{C}\hat{C}^T)\}. \tag{65}$$

Since norms are distinct and sorted, this implies $X_i = \hat{X}_i$ for all $i$. By Theorem 6.4 in (Blum-Smith et al., 2024), the remaining terms satisfy $CZ_i^T = \hat{C}\hat{Z}_i^T$ and $CC^T = \hat{C}\hat{C}^T$ only if there exists $t \in O(C)$ such that $tZ_i = \hat{Z}_i$. As norms are sorted, $t$ must be the identity, proving expressivity. $\qquad\square$

Therefore, there exists a parameterization that the aggregator can differentiate any two different inputs that orthogonal transformations cannot transform them to each other except some rare cases that the equivariant features are very symmetric that the representative orientation degenerates (for example, to zero). These cases are important for other domains using equivariant models, like 3D molecule, where symmetric structures are common and affects molecular properties significantly. However, our equivariant representations are initialized from noise, so the symmetric is very rare.

## G.2 S(C)-EQUIVARIANT AGGREGATOR

Following (Maron et al., 2020), we propose permutation equivariant aggregators. This aggregator is permutation-equivariant to node permutations ($S_n$) and noise channel permutations ($S_C$).

The aggregator follows four main steps:

1. **Identifying Noise Channels**: Apply a DeepSet $\psi : \mathbb{R}^{n \times L} \to \mathbb{R}^{L_0}$ to generate unique identifiers for each noise channel:
$$Z_{i,:,c}^1 = Z_{i,:,c} \| \psi(Z_{:,:,c}). \tag{66}$$

2. **Node Encoding**: Combine noise and invariant features via a DeepSet $\phi : \mathbb{R}^{(L+L_0) \times C} \to \mathbb{R}^{L_1}$:
$$X_i^0 = \phi(Z_i^1) \| X_i. \tag{67}$$

3. **Set Encoding**: Aggregate node features with a DeepSet $\varphi : \mathbb{R}^{k \times (d+L_1)} \to \mathbb{R}^{d_1}$:
$$X^1 = \varphi(X^0). \tag{68}$$

4. **Generating Equivariant Outputs**: Use MLPs $g$ and $h$ to produce outputs:
$$X_i^2 = g(X^1 \| X_i^0), \quad Z_{i,:,c}^2 = h(X^1 \| X_i^0 \| Z_{i,:,c}^1). \tag{69}$$

Each step of the aggregator is efficient, with a time and space complexity of $\Theta(k)$.

Let AGGR denote our aggregator. AGGR guarantees equivariance to node and noise channel permutations. Formally:

**Proposition G.3.** *For any parameterization of $\psi, \phi, \varphi, g, h$, features $X \in \mathbb{R}^{k \times d}$, noise features $Z \in \mathbb{R}^{k \times L \times C}$, and permutations $P_1 \in S_k$ for node, $P_2 \in S_C$ for noise channel, if $X', Z' = AGGR(X, Z)$, then:*

$$P_1(X'), P_2(P_1(Z')) = AGGR(P_1(X), P_2(P_1(Z))). \tag{70}$$

*Proof.* Note that DeepSet model is permutation invariant to permutation on the dimension it aggregates. Moreover, if operator act individually on some dimension, the operator is also equivariant to permutation on the dimension.

Therefore, when $Z \to P_2(P_1(Z))$, $\psi(Z_{:,:,k}) = \psi(Z_{:,:,P_2^{-1}(k)})$, so $Z^1 \to P_2(P_1(Z^1))$.

With $Z^1 \to P_2(P_1(Z^1))$, and $x \to P_1(x)$, $x^0 \to P_1(x^0)$, so $X^1 \to X^1$, $X^2 \to P_1(X^2)$, and $Z^2 \to P_2(P_1(Z^2))$. $\qquad\square$

Under mild conditions, AGGR can approximate any equivariant continuous function. Formally:

**Proposition G.4.** *Given a compact set $U \subseteq \mathbb{R}^{k \times d} \times \mathbb{R}^{k \times L \times C}$ that for all each channel of noise has a different elements multiset, AGGR is a universal approximator of continuous $S_k \times S_C$-equivariant functions on $U$.*

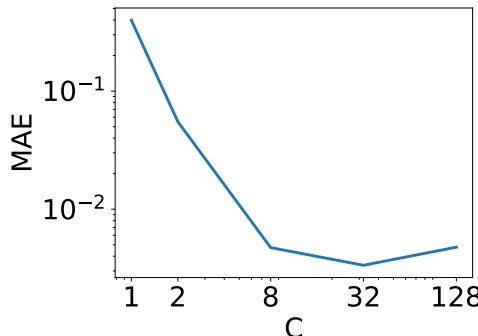

Figure 2: Test MAE of triangle counts on subgraph counting task for ENGNN-O with different noise space dimension $C$.

*Proof.* Following Segol & Lipman (2020), we first show that the set encoding $X^1$ in our aggregator can approximate any invariant function first. As permutation equivariant function can be expressed as a elementwise transformation conditioned by the invariant function, we can easily approximate any equivariant output.

According to Stone-Weierstrass theorem, to prove the universality of set encoding $X^1$ is equivalent to that our aggregator can differentiate any two input set of invariant and equivariant features with some parameterization.

Let all DeepSet and MLPs in our aggregator be injective. Given two set of features $X \in \mathbb{R}^{k \times d}, Z \in \mathbb{R}^{k \times L \times C}$ and $X' \in \mathbb{R}^{k \times d}, Z' \in \mathbb{R}^{k \times L \times C}$, if $X^1 = X'^1$, then,

- As $\varphi$ is injective, $\exists P_1 \in S_k, P_1(X^0) = X'^0$.
- As $P_1(X^0) = X'^0$, $P_1(X) = X'$. Moreover, $P_1(\phi(Z^1)) = \phi(Z'^1)$.
- As $P_1(\phi(Z^1)) = \phi(Z'^1)$, for each row $\phi(Z^1)_i = \phi(Z'^1)_{P_1(i)}$, $\exists P_{2i} \in S_C, Z_i^1 = P_{2i}(Z'^1_{P_1(i)})$. The permutation of noise channel may be different for each row, but each noise channel is assigned with unique column label, so $P_{2i}$ are all equal to $P_2$. Therefore, $P_2(P_1(Z^1)) = Z'^1$.
- So $P_2(P_1(Z)) = Z'$,

Therefore, the invariant representation is universal. $\qquad\square$

### G.3 AGGREGATOR FOR ABLATION.

For ablation model MPNN, we use aggregator using invariant features only as follows.

1. With MLP $f : \mathbb{R}^d \to \mathbb{R}^{d'}$ and $g' = \mathbb{R}^{d'} \to \mathbb{R}^d$, $X' = g \sum_{i=1}^n f(x_i)$.
2. $Z' = 0$.

For ablation model NMPNN, we use aggregator as follows.

1. With MLP $f : \mathbb{R}^{d+CL} \to \mathbb{R}^{d'}$ , $C = \sum_{i=1}^n f(x_i \| Z_i)$.
2. With MLP $f : \mathbb{R}^{d'} \to \mathbb{R}^d$ and $h : \mathbb{R}^{d'} \to \mathbb{R}^{CL}$, $Z' = h(C)$, and $X' = f(C)$.

## H  NOISE DIMENSION ABLATION

We conduct ablation study on triangle counting task. The results are shown in Figure 2. As shown in the Figure, with $C = 1$, the expressivity is low and loss is high. From $C = 1$ to $C = 32$, test loss decrease as $C$ increase, as the expressivity increases. However, from $C = 32$ to $C = 128$, test loss increases, as the noise space gets larger and the sample complexity increases, leading to larger generalization error.

Table 9: Roc-auc score ↑ with uncertainty of ENGNN and rGIN on synthetic.

| dataset | TRI(N) | TRI(X) | LCC(N) | LCC(X) | MDS(N) | MDS(X) |
|---------|--------|--------|--------|--------|--------|--------|
| GINs | $0.500_{\pm 0.000}$ | $0.500_{\pm 0.000}$ | $0.500_{\pm 0.000}$ | $0.500_{\pm 0.000}$ | $0.500_{\pm 0.000}$ | $0.500_{\pm 0.000}$ |
| rGINs | $0.908_{\pm 0.013}$ | $0.926_{\pm 0.014}$ | $0.811_{\pm 0.005}$ | $0.852_{\pm 0.007}$ | $0.807_{\pm 0.009}$ | $0.810_{\pm 0.008}$ |
| MPNN | $0.500_{\pm 0.000}$ | $0.500_{\pm 0.000}$ | $0.500_{\pm 0.000}$ | $0.500_{\pm 0.000}$ | $0.500_{\pm 0.000}$ | $0.500_{\pm 0.000}$ |
| NMPNN | $1.000_{\pm 0.000}$ | $1.000_{\pm 0.000}$ | $1.000_{\pm 0.000}$ | $1.000_{\pm 0.000}$ | $0.933_{\pm 0.002}$ | $0.932_{\pm 0.001}$ |
| ENGNN-P | $\mathbf{1.000}_{\pm 0.000}$ | $\mathbf{1.000}_{\pm 0.000}$ | $\mathbf{1.000}_{\pm 0.000}$ | $\mathbf{1.000}_{\pm 0.000}$ | $0.936_{\pm 0.003}$ | $0.934_{\pm 0.004}$ |
| ENGNN-O | $\mathbf{1.000}_{\pm 0.000}$ | $\mathbf{1.000}_{\pm 0.000}$ | $\mathbf{1.000}_{\pm 0.000}$ | $\mathbf{1.000}_{\pm 0.000}$ | $\mathbf{0.938}_{\pm \mathbf{0.006}}$ | $\mathbf{0.939}_{\pm \mathbf{0.002}}$ |

Table 10: Comparison between our ENGNN and more noise GNN methods.

| Dataset | EXP | CEXP | TRI(N) | TRI(X) | MUTAG | NCI1 | PROTEINS |
|---------|-----|------|--------|--------|-------|------|----------|
| CLIP | 0.99±0.04 | 0.99±0.02 | 0.99±0.00 | 0.81±0.05 | 0.85± 0.09 | 0.81±0.01 | 0.65±0.05 |
| RP | 0.96±0.02 | 0.97±0.02 | 0.99±0.00 | 0.82±0.03 | 0.86± 0.07 | 0.81±0.01 | 0.74±0.04 |
| IRNI | 0.99±0.04 | 0.95±0.14 | 0.99±0.01 | 0.73±0.04 | 0.85±0.05 | 0.82±0.02 | 0.75±0.04 |
| GPSE | 0.74±0.01 | 0.75±0.02 | 0.96± 0.01 | 0.90±0.09 | N/A | N/A | 0.835±0.001 |
| ENGNN-P | **1.00±0.00** | **1.00±0.00** | **1.00±0.00** | **1.00±0.00** | **0.990± 0.019** | **0.897± 0.013** | **0.837± 0.027** |
| ENGNN-O | **1.00±0.00** | **1.00±0.00** | **1.00±0.00** | **1.00±0.00** | **0.990± 0.014** | **0.902± 0.022** | **0.843± 0.028** |

# I EXTRA EXPERIMENTS

We show the uncertainty on synthetic datasets of rGIN (Sato et al., 2021) in Table 9. The standard is small compared with the score gap between baseline rGIN and our ENGNN, validating the effectiveness of our model.

We compare our ENGNN with more noise methods in Table 10.

Following Pellizzoni et al. (2024), we verify the generalization capacity in limited number of training sample case and evaluate the train-test performance gap. Baseline are from Pellizzoni et al. (2024). Our ENGNN achieves smaller train-test gap on all training set size (Table 11) and better performance and smaller train-test gap on TU datasets (Table 12).

Table 11: Train-test performance gap with limited number of training samples on 3-regular graph dataset

| training size | RNI | RP | Tinhofer | ENGNN-O | ENGNN-P |
|---|---|---|---|---|---|
| 1 | 0.497± 0.001 | 0.502± 0.002 | 0.413± 0.005 | 0.0± 0.0 | 0.0± 0.0 |
| 10 | 0.489± 0.001 | 0.445± 0.003 | 0.213± 0.002 | 0.008± 0.006 | 0.0± 0.0 |
| 100 | 0.310± 0.204 | 0.421± 0.002 | 0.002± 0.0 | -0.007± 0.005 | 0.0± 0.0 |
| 1000 | 0.394± 0.012 | 0.298± 0.059 | 0.0± 0.0 | -0.003± 0.001 | 0.0± 0.0 |

Table 12: Train-test performance gap on TU datasets. Test mean test AUC. Diff means train-test AUC gap.

| | NCI1 | | MUTAG | | IMDB | | COLLAB | |
|---|---|---|---|---|---|---|---|---|
| | Test | Diff | Test | Diff | Test | Diff | Test | Diff |
| None | 81.8±1.4 | 18.1±1.5 | 81.5±1.3 | 18.4±1.3 | 71.4±3.9 | 2.2±4.4 | 75.3±1.1 | 0.0±1.0 |
| RP | 67.4±1.2 | 32.6±1.2 | 71.2±1.0 | 28.8±1.0 | 63.8±2.3 | 34.6±2.3 | 75.1±2.3 | 18.1±3.1 |
| RNI | 68.0±1.0 | 32.0±1.0 | 72.7±2.4 | 27.3±2.4 | 63.6±8.8 | 36.4±8.8 | 72.1±2.8 | 19.6±4.9 |
| Tinhofer | 72.9±3.0 | 26.9±3.0 | 73.1±2.3 | 26.9±2.3 | 68.6±3.1 | 20.5±3.1 | 75.2±1.7 | 19.5±1.2 |
| Tinhoferw | 81.8±2.3 | 18.1±2.4 | 81.2±1.7 | 18.8±1.7 | 69.4±3.4 | 19.6±3.2 | 80.8±1.9 | 12.2±2.1 |
| LPE | 76.4±1.9 | 23.5±1.9 | 75.4±2.0 | 24.6±2.0 | 68.4±3.3 | 20.7±3.3 | 75.8±1.8 | 19.9±2.3 |
| ENGNN-P | 84.8± 1.5 | 15.1± 1.5 | 91.7± 6.2 | 8.3± 6.2 | 78.7± 1.2 | 4.3± 1.2 | 81.9±1.9 | 1.3± 1.5 |
| ENGNN-O | 85.0± 1.3 | 14.9± 1.3 | 94.4± 5.5 | 5.6± 5.5 | 79.5± 0.5 | 4.1± 0.5 | 81.9± 0.5 | -1.5±0.5 |

