# OpenReview forum: "Expressive Graph Neural Networks via Equivariant Use of Noise"
_ICLR.cc/2026/Conference — Submitted to ICLR 2026_

### Official Review · Reviewer_yQhu · 2025-10-14

**Soundness:** 1
**Presentation:** 4
**Contribution:** 2
**Rating:** 2
**Confidence:** 4

**Summary:**

This paper proposes adding noise to graph data, and designing a GNN which respects some symmetries of the noise, to achieve universal approximation and improve sample complexity with respect to past methods which injected noise to the data.

**Strengths:**

The paper is well written and the method seems to perform well on benchmark problems.

**Weaknesses:**

I believe that the analysis proposed in the paper is fundamentally flawed.

In general, the treatment of the noise channels in the analysis is strange. The paper does not treat the noise as a random variable in the analysis. The paper simply treats the “noise” as deterministic channels of the node features. The authors even assume that the ground-truth function that maps input to (ground-truth) outputs explicitly depends on the value of the “noise.” This does not model appropriately the setting discussed in this paper, where noise is injected into the features “by the user” as part of the computation of the GNN. The authors somehow assume that noise is a ground-truth part of the data and the task, which is not correct.

For example, in the way the authors model the setting, if given a datapoint G you once realize the noise as Z and then you realize the noise as Z’, the ground-truth output in these two cases should be different. But this breaks how the data distribution should work. The noise Z is only something the user does to the data as a computational trick, it is not part of the data and should not affect the task/ground-truth target. This treatment of Z even breaks the very definition of noise in signal processing. Noise is not something that affects the ground-truth. The ground-truth depends only on the clean data.


Specifically, in Line 212 the authors write that h depends on Z. Z is a random noise that the “user” adds to the data to implement some computational trick. This noise has nothing to do with the actual ground-truth data and with the problem/task. This noise is not given as part of the dataset. Hence, h can only depend on G. Allowing the estimator of h to depend on Z is only a computational trick.


Moreover, the only theorem I could find regarding universal approximation is Theorem 4.5, where the universal approximation property is assumed on the aggregator, without explaining what this means. When reading Appendix G, the reader then discovers that here again the analysis assumes that the “noise” channels are just some deterministic ground-truth channels of the input.  Moreover, the universal approximation is formulated for target functions that respect the symmetry of the “noise.” This is again a strange setting. The target function should not depend on the noise at all, as this noise was injected by the “user.” Hence the target function should be trivially invariant to any symmetry of the noise. Even if somehow the authors think that the ground-truth depends on the user-injected noise, how do the authors justify the assumption that this ground-truth function should respect a given symmetry? This is not discussed anywhere.

The authors should reconsider their analysis, and seek alternative theoretical routes for explaining the success of their method.

**Questions:**

1. What is the motivation behind assuming that the ground-truth target depends on user-injected noise?

2. Why not use a simpler solution to solve the sample complexity -- simply take less noise channels, e.g., one channel? The paper should explain why your solution is better than the single noise channel solution, which is much simpler and also solves the sample complexity problem. Note that using only one noise channel gives you all the theoretical benefits that using multiple noise channels gives.

3. What is the motivation behind using specific symmetries for the noise and not others? Can you theoretically base the choice of the symmetry?

---

### Official Review · Reviewer_XXCW · 2025-10-17

**Soundness:** 1
**Presentation:** 3
**Contribution:** 2
**Rating:** 2
**Confidence:** 5

**Summary:**

In this paper, the authors focus on reconciling universal expressivity and practical generalization in GNNs. They propose ENGNN, which injects nodewise noise and processes it equivariantly to a chosen transformation group, such as orthogonal or permutation, so outputs are invariant to noise transformations while retaining expressivity. ENGNN uses a two-stream message passing design with an invariant stream for original features and an equivariant stream for noise, combined via specially designed equivariant aggregators called ENGNN-O and ENGNN-P. Extensive experiments are conducted on graph, node, link, and subgraph benchmarks.

**Strengths:**

Strengths:

- The authors clearly articulates the motivation behind introducing equivariant noise into GNNs, which is to reconcile the trade-off between theoretical expressivity and empirical generalization.
- The experiments are conducted on a wide range of different tasks and benchmarks.

- The authors provide theoretical support, connecting the notion of noise equivariance to reduced sample complexity using PAC learning theory.

**Weaknesses:**

Weaknesses:

- The theoretical contributions are limited. For example, Theorem 4.1 on sample-complexity reduction and Theorem 4.5 on universal expressivity primarily extend existing frameworks. Specifically, the generalization bound closely follows invariance-based analyses [1,2], and the universality claim builds on prior random-feature arguments [3,4], where distinct random features already suffice to guarantee universality. Additionally, Theorem 4.5 assumes each node has distinct random features, which are hard to guarantee in finite-dimensional, finite-precision practice.
- The proposed equivariant aggregator in Section 4.2 is described only conceptually without an explicit mathematical definition, which is hard to follow. After checking the Appendix G, it is still unclear that how the aggregator is incorporated into the ENGNN layers and how equivariance is enforced during message passing.
- **Most importantly,** in the experimental section, the authors evaluate ENGNN across a broad set of benchmarks, but two issues weaken the evidence: (1) The baselines are not up to date, which limits the fairness of the comparisons: for example, in Table 4 the most recent methods are GPRGNN and BernNet (2021), and in the subgraph tasks the newest baseline is GLASS (2022). Several stronger, more recent approaches [5,6,7, etc] are missing; (2) Across many tasks ENGNN does not achieve state-of-the-art (SOTA) performance, as shown in Tables 5 and 6. Due to the compared baselines are out of date, the reported SOTA performance are not convinced.

Although the paper is thoughtfully designed and well motivated, the weaknesses above, particularly in the experimental evaluation, leave me unconvinced about the practical utility of the proposed method.

[1] On the Sample Complexity of Learning under Invariance and Geometric Stability.

[2] The Exact Sample Complexity Gain from Invariances for Kernel Regression.

[3] The Surprising Power of Graph Neural Networks with Random Node Initialization.

[4] Random Features Strengthen Graph Neural Networks.

[5] An Efficient Subgraph GNN with Provable Substructure Counting Power.

[6] Towards Dynamic Message Passing on Graphs.

[7] Pure Message Passing Can Estimate Common Neighbor for Link Prediction

**Questions:**

Questions:

- What are the specific mathematical form of $AGGR_1$ and $AGGR_2$ in Eq. (4)?
- How does the complexity of the aggregator scale with noise dimension $C$ and transformation group $T$?
- Does the model remain universal if noise is fixed rather than resampled each epoch?

---

### Official Review · Reviewer_Ys8m · 2025-10-26

**Soundness:** 3
**Presentation:** 3
**Contribution:** 3
**Rating:** 4
**Confidence:** 4

**Summary:**

This paper introduces Equivariant Noise Graph Neural Networks (ENGNNs), a theoretically and empirically grounded framework for enhancing the expressivity of Graph Neural Networks (GNNs) while maintaining generalization and scalability. Traditional GNNs (i.e., MPNNs) struggle to distinguish non-isomorphic graphs, while previous approaches using random node noise improve expressivity but harm generalization. ENGNN resolves this by making the model equivariant to transformations in the noise space (i.e., orthogonal transformations or channel permutations), thereby reducing sample complexity. The authors prove that ENGNNs are both universally expressive for graph, subgraph, and node-level tasks, and invariant to chosen noise transformations. They implement two variants -  ENGNN-O (orthogonal-equivariant) and ENGNN-P (permutation-equivariant) - and demonstrate state-of-the-art or competitive results across synthetic and real-world benchmarks (graph property prediction, node classification, link prediction, and subgraph tasks), all while preserving the linear scalability of standard MPNNs.

**Strengths:**

1. Introduces a novel theoretical connection between equivariance in noise space and sample complexity reduction, filling a key gap between highly expressive but unscalable GNNs (i.e., higher-order WL) and practical but under-expressive ones (i.e., MPNNs). The idea of treating random noise as an equivariant latent variable rather than mere augmentation is conceptually elegant and theoretically sound.
2. It establishes a new design principle for expressive GNNs, showing that equivariance in the noise space can serve as a bridge between expressivity and generalization while enabling both theoretical universality and linear empirical scalability.
3. Empirical results are comprehensive - spanning graph-, node-, link-, and subgraph-level benchmarks - showing consistent improvements over both naive noise-based GNNs and high-order baselines.

**Weaknesses:**

1. **Limited Large-Scale Evaluation:** Despite linear complexity claims, the model is not tested on large-scale benchmarks like OGB-LSC, PPA, or MolPCQM4Mv2. Such experiments are critical to verify the real-world scalability and memory efficiency of ENGNN beyond synthetic and TU datasets.
2. **Proof Rigor and Assumptions:** The PAC-learning derivation (Theorem 4.1) assumes bounded Lipschitz constants and compactness, which may not hold in ReLU-activated MLPs.
3. **Ablation Depth:** Ablations on noise type, transformation group, and dimensionality are limited. A table showing how performance scales with these hyperparameters would greatly strengthen empirical understanding.
4. **Interpretability:** While theoretically elegant, the effect of equivariant noise is somewhat abstract. Visualizations showing how node embeddings differ between noise and equivariant noise cases would improve intuition.

**Questions:**

1. **Clarification on Noise Transformation Groups (Theorem 4.4 \& 4.5):** While the proofs rely on equivariance to a transformation group $T$, the paper lacks explicit definition of the group composition and closure properties required for invariance proofs. Could the authors specify whether $T$ is compact (as assumed in PAC-learning theory for sample complexity bounds)?
2. **Practical Implementation of Equivariant Aggregators*:* Appendix G is said to define orthogonal- and permutation-equivariant aggregators, but the paper does not clearly describe how these are parameterized or trained.
3. **Noise Resampling and Stability:** The noise $Z$ is resampled at each forward pass, similar to dropout. How does this stochasticity interact with the equivariance constraint? Does resampling affect consistency during inference?
4. **Scalability Beyond TU/ZINC Datasets:** Although ENGNN shows linear theoretical complexity, most datasets used are small- to medium-scale. Could the authors provide results on larger graphs (i.e., OGB-LSC, OGBG-PPA, or PCQM4Mv2)? Even a memory/runtime table would better establish the claimed scalability.
5. **Generalization Bound (Theorem 4.1) - Assumption Check:** The PAC-based bound assumes Lipschitz continuity in both $G$ and $Z$. Is this assumption realistic for MPNNs with ReLU activations? If not, could the authors provide empirical or relaxed justifications?
6. **Ablation on Noise Dimensionality and Transformation Type:** How sensitive is performance to the number of noise channels $C$? Does orthogonal equivariance outperform permutation equivariance across tasks, or vice versa?
7. **Expressivity Validation:** Theoretical expressivity (Theorem 4.5) assumes distinct noise vectors per node. In practice, when noise dimensions are limited, how often do collisions reduce discriminative power?

---

### Official Review · Reviewer_UpeX · 2025-11-03

**Soundness:** 2
**Presentation:** 1
**Contribution:** 2
**Rating:** 4
**Confidence:** 3

**Summary:**

The paper introduces Equivariant Noise GNNs (ENGNNs), a framework that enhances the expressivity of message-passing GNNs by injecting random node-wise noise while enforcing equivariance to noise transformations (orthogonal and channel permutations). The central premise is that naive noise features can recover universal expressivity but hurt generalization by dramatically expanding the hypothesis space. The authors propose a two-stream architecture (invariant stream for standard node features, equivariant stream for random noise) and an equivariant aggregator that jointly preserves symmetry and mixes information.

They provide:

- A PAC-learning-based sample complexity analysis showing that invariance/equivariance to noise reduces the covering number of the noise space, tightening generalization bounds.

- Formal proofs that ENGNN is invariant to the chosen noise transformations and achieves universal expressivity for graph-, node-, link-, and subgraph-level tasks, contingent on universally expressive aggregators.

- Comprehensive experiments demonstrating strong performance across synthetic and real-world datasets for graph, node, link, and subgraph tasks, often matching more expensive high-order/subgraph GNNs while remaining scalable.

**Strengths:**

- The idea to enforce equivariance on auxiliary random noise is fresh and well-argued. It reframes noise from “node individualization” to “symmetry-controlled augmentation,” leveraging group actions to tame sample complexity.

- Theory: The PAC-style argument is clear and connects symmetry to smaller covering numbers, aligning with broader invariance-generalization literature. Propositions showing factorial reductions under channel permutations are compelling, and the invariance + expressivity theorems are stated cleanly. The aggregator constructions are grounded in prior equivariant set-function theory yet adapted to the noise setting.

- Empirics: Broad coverage across tasks (graph property prediction, subgraph counting, node classification, link prediction) with comparisons to both naive-noise MPNNs and more expressive/high-cost baselines.

**Weaknesses:**

- The presentation is not clear and is missing important details.
  - The paper mentions generalization multiple times and claims "NMPNN ... compromises generalization .." without formally defining it. If the authors mean "gap between training and test error", I don't see how NMPNN compromises generalization from the figures
  - The paper is missing related works on GNN generalization
  - On L225: "h ∈ H are $C_G$-Lipschitz in $\mathcal{G}$", to define $C_G$-Lipschitz properly, the authors should describe the distance metric used in $\mathcal{G}$.
  - Eqn 4 seems incorrectly presented, the formula doesn't match the description down below.
  - The invariant and equivariant feature of ENGNN should be the key contribution but the paper only mentioned them lightly in the main text, it is unclear to me how and why it is invariant/equivariant.

- Some arguments are not well-supported, e.g. L235: "For an n × C-dimensional boolean noise space, NZ can be as large as $2^{nC}$ , ..This explains why noise leads to poor generalization" If this is true, larger n should also lead to poor generalization which contradicts other literature that generalization improves on larger graphs.

- Assumptions in expressivity proofs: Theoretical universal expressivity depends on distinct random features and universally expressive aggregators; the paper asserts aggregators are universally expressive “under mild conditions.” A more explicit statement of these conditions, with pointer to parameter ranges or regularity assumptions, would help reproducibility and clarity.

- Empirics.ENGNN trails some baselines such as subgraph-based models. The discussion attributes this to subgraph inductive biases. A deeper analysis (e.g., adding task-specific structural cues to ENGNN or measuring when ENGNN underperforms due to symmetry constraints) would be informative.

**Questions:**

- How is ENGNN permutation-invariant? By adding random noise the node permutation would lead to noise being injected on different nodes right? The other noise GNN works only claims "invariant by expectation", is this the same case here?

- On the PAC bound: Can you provide more concrete numeric or asymptotic estimates of NZ,T for typical settings (e.g., Gaussian noise with finite precision, realistic C, n) to contextualize the practical magnitude of the generalization gain? A small table or simulation would help.

- On aggregator universality: The universal approximation claims depend on “mild conditions.” Could you specify these precisely?

- On noise resampling: You resample noise i.i.d. per forward pass (as in Abboud et al.). Have you tried fixed-noise training vs resampled-noise training and measured variance/generalization differences? If equivariance theoretically reduces effective hypothesis space, might fixed-noise suffice and improve training stability?

- On group choices: Why focus on O(C) and S(C)? Have you explored other groups (e.g., scaling groups, signed permutations, block-permutations matching node types)? Could hybrid groups further reduce NZ,T without harming expressivity?

---

### Meta-Review · Area_Chair_uMf3 · 2025-12-23

**Summary:**

This paper proposes Equivariant Noise GNNs (ENGNNs), a framework that injects random noise into node features while enforcing equivariance to noise transformations, aiming to achieve universal expressivity without sacrificing generalization. The core idea—treating noise as a symmetry-controlled augmentation rather than naive individualization—is elegant and well-motivated, and the empirical results are reasonably comprehensive. However, Reviewer yQhu, who has deep expertise in this area, raised fundamental concerns about the theoretical analysis. Other reviewers noted presentation issues and missing experimental comparisons. In the absence of a rebuttal addressing these theoretical concerns, the paper cannot be accepted in its current form. That said, the underlying idea is appealing, and the empirical performance suggests the approach has merit; I would encourage the authors to carefully revisit the theoretical formulation and resubmit to a future venue.

**Reviewer Concerns:**

no rebuttal

**Reviewer Scores:**

no rebuttal

---

### Decision · Program_Chairs · 2026-01-26

Reject